# Hybrid TiO$_2$/Al$_2$O$_3$ nanolayer overcoating enhances dental lithium disilicate glass-ceramics acid resistance and surface mechanical properties

Xiaoxuan Zhang [1,6], Bin Zhang [2,6] ✉, Fanchun Meng[2], Xiangyu Zhao[1], Ruilin Liu[2], Meixian Liu[1], Huifei Li[1], Yuan Zhang[1], Feng Wu[1], Jin Bai[2], Zhuoqun Yan[3], Xiuyun Ren[1], Yong Qin [4] ✉ & Xing Wang [1,5] ✉

Lithium disilicate glass-ceramics are increasingly utilized in dental applications due to their exceptional optical and aesthetic properties. However, lithium disilicate glass-ceramics possess a mixed crystalline and amorphous structure with variable resistance to external corrosion. Exposure of lithium disilicate glass-ceramics to acidic conditions in the oral cavity leads to pore formation, and the risk of cracking increases with long-term wear. Herein, we employed atomic layer deposition to precisely coat the TiO$_2$/Al$_2$O$_3$ hybrid nanolayer on the surface of lithium disilicate glass-ceramics to enhance acid resistance and surface mechanical properties. The TiO$_2$ species play crucial roles in improving acid resistance, and the Al$_2$O$_3$ species, atomically bonded to the TiO$_2$ species in the hybrid layer, promote interfacial interactions. Moreover, the TiO$_2$/Al$_2$O$_3$ hybrid nanolayer shows no cytotoxicity while maintaining the aesthetics of lithium disilicate glass-ceramic restorations, thus offering broad prospects for oral restoration applications.

With the continuous rise in aesthetic demands, there is an annual increase in the demand for all-ceramic dental restorations, including crowns and veneers. According to a market report from Fortune Business Insights, the global dental crown market value was $1.8 billion in 2018 and is expected to reach $2.9 billion by 2026[1,2]. Lithium disilicate glass-ceramic (LDGC) is widely used in clinical practice due to its unique aesthetic advantages and cutting performance[3–5]. The flagship products, for example, IPS e.max CAD with a flexural strength of up to 360 MPa[6,7], have been fixtures in the dental market for over 15 years[8]. However, clinical research has shown a failure rate of 4.0–13.6% when using LDGC for crown restoration[9–11]. Research efforts are predominantly concentrated on augmenting the mechanical strength of

the LDGC. Typical solutions include optimizing the sintering time and adding zirconia components[12–16]. For instance, the machinable zirconia-reinforced lithium silicate was developed under the models of VITA Suprinity and Celtra Duo, respectively[17]. Their flexural strength can reach 451.4 MPa[18]. Although the overall strength is improved, excessive addition of zirconia affects the transparency of the ceramic and even increases the risk of crack generation during the cutting process[19–22]. In addition, the aging of ceramics in an acidic environment poses a significant challenge, hindering the long-term application of prostheses. Excessive consumption of carbonated beverages, bacterial colonization, and gastric acid reflux due to disease can all contribute to an acidic oral environment. LDGC has a mixed structure of crystalline

[1]Shanxi Medical University School and Hospital of Stomatology, Taiyuan, China. [2]State Key Laboratory of Coal Conversion, Institute of Coal Chemistry, Chinese Academy of Sciences, Taiyuan, China. [3]Innovation Center, Upcera Co. Ltd., Shenyang, China. [4]College of Materials Science and Engineering, Qingdao University of Science and Technology, Qingdao, China. [5]Shanxi Province Key Laboratory of Oral Diseases Prevention and New Materials, Taiyuan, China. [6]These authors contributed equally: Xiaoxuan Zhang, Bin Zhang. ✉e-mail: zhangbin2009@sxicc.ac.cn; qinyong@qust.edu.cn; wangxing@sxmu.edu.cn

and amorphous components. Various components exhibit different dissolution rates in acidic solutions, thereby giving rise to structural inhomogeneity[12,23–27]. Overcoming the limitations of existing approaches requires strategies to enhance acid resistance while maintaining proper surface mechanical properties in ceramic crowns.

A surface coating strategy is needed to protect the ceramic surface, reduce the risk of crack generation, meet oral-use requirements, and ensure long-term durability. Three key factors must be taken into account. Firstly, the tooth surface has intricate groove structures, requiring a high-density, conformal film to prevent contact between the ceramic substrate and the complex oral environment. Secondly, as oral aesthetic prosthesis materials, glass ceramics must maintain transparency without compromising their aesthetic properties, making nanoscale thickness control imperative. Thirdly, modified films' surface mechanical properties should meet the oral-use requirements for compatibility with existing teeth. However, previous modification schemes, such as surface grit-blasting and acid-etching, have been found to result in stress concentration and increased roughness[28]. Additionally, bioactive membranes exhibit poor mechanical properties that hinder their ability to withstand forces[29], consequently leading to a diminished anti-corrosion effect due to the easy contact between the ceramic substrate and acidic solution. Furthermore, traditional glazing methods cannot effectively control the thickness and require high-temperature sintering, which is detrimental to ceramic crystals[30]. Due to these technical limitations, no attempts have been made to address acid resistance and controlled surface mechanical properties in glass-ceramic crowns through surface modification.

Atomic layer deposition (ALD) is a vapor phase technique that enables precise film thickness control at the atomic level, resulting in conformal overcoating on complex substrates, even at relatively low temperatures[31]. ALD-deposited nanofilms enhance the substrates' physical and chemical properties depending on the deposition types. For instance, depositing sub-10 nm thick ALD $TiO_2$ films onto silk fiber surfaces improves their anti-fouling properties[32]. Additionally, $Al_2O_3$ deposition leads to films with increased hardness and modulus[33]. $TiO_2$ capping layers (-200 Å thick) on $Al_2O_3$ adhesion layers protect copper for approximately 80 days in water at 90 °C[34]. ALD growth is achieved through the formation of chemical bonds between films and substrate[35,36]. Since lithium silicate substrates have many functional groups for ALD deposition, the deposition of nano oxide films onto LDGC crown surfaces would result in a highly stable and compact film by forming strong chemical bonds. Moreover, the uniform nanometer-thick film is expected to enhance surface corrosion resistance and tailor the surface mechanical properties without affecting the original color of LDGC.

In this work, ALD was utilized to deposit a $TiO_2/Al_2O_3$ hybrid nanolayer as an overcoat on LDGC surfaces, considering the corrosion resistance of $TiO_2$ and the isotropic nature of $Al_2O_3$. We compared the properties of LDGC overcoated with a $TiO_2/Al_2O_3$ hybrid nanolayer against those overcoated with an $Al_2O_3$ nanolayer, a $TiO_2$ nanolayer, and a $TiO_2$-$Al_2O_3$ (or $Al_2O_3$-$TiO_2$) double nanolayer. We focused on evaluating how film composition and film-substrate interaction impact both acid resistance and mechanical properties of LDGC. Our findings demonstrate successful overcoating of the highly transparent $TiO_2/Al_2O_3$ hybrid nanolayer onto LDGC without compromising its aesthetic qualities. This approach significantly enhances the surface hardness and does not affect the modulus of the ceramics. Furthermore, our results reveal that the LDGC coated with $TiO_2/Al_2O_3$ hybrid nanolayer exhibits superior acid resistance compared to the uncoated LDGC or LDGC coated with $Al_2O_3$ nanolayer, even at pH 2.5. Additionally, it demonstrates remarkable mechanical stability against surface nanoscratch compared to the LDGC coated with a $TiO_2$ nanolayer or a $TiO_2$-$Al_2O_3$ (or $TiO_2$-$Al_2O_3$ with reverse structure) double layer. This work highlights the potential for composite nanolayers in ceramic applications while providing insights into dental crown materials and other ceramic implants.

## Results

### Characterization of LDGC with coated nanolayer

The surface of LDGC was coated with a $TiO_2/Al_2O_3$ hybrid nanolayer, denoted as $TiO_2/Al_2O_3@LDGC$, through sequential alternate ALD of $TiO_2$ and $Al_2O_3$ at 150 °C under vacuum conditions (Fig. 1a)[37,38]. This process involved cyclic pulsing of titanium tetraisopropoxide, water, trimethyl aluminum, and water for 150 cycles. For comparison, LDGC samples were also coated with individual layers of $TiO_2$ or $Al_2O_3$ as well as $TiO_2$-$Al_2O_3$ (or $Al_2O_3$-$TiO_2$) double nanolayer, and the samples were denoted as $TiO_2@LDGC$, $Al_2O_3@LDGC$, $TiO_2$-$Al_2O_3@LDGC$, and $Al_2O_3$-$TiO_2@LDGC$, respectively.

Scanning electron microscope (SEM) images revealed the surface morphology of all samples (Fig. 1b and Supplementary Fig. 1). After overcoating via ALD, the surface film of LDGC exhibits a relatively uniform distribution state. Enhanced smoothness with reduced defects was observed for $TiO_2/Al_2O_3@LDGC$. Energy dispersive X-ray spectroscopy (EDS) analysis revealed a uniform distribution of deposited elements on the surface of the LDGC (Supplementary Fig. 2). A cross-section of $TiO_2/Al_2O_3@LDGC$ (Fig. 1c) prepared using focused ion beam-scanning electron microscope (FIB-SEM) allowed us to evaluate the cross-section of the $TiO_2/Al_2O_3$ hybrid nanolayer, which exhibited a thickness of only 33 nm (Fig. 1d), significantly below the perceptible limit for human teeth. Furthermore, Fig. 1e shows the homogeneous dispersion of Ti and Al elements but no observed lattice fringe for the $TiO_2/Al_2O_3$ hybrid nanolayer, indicating the formation of an amorphous hybrid dense film devoid of pores or nanocrystals. This observation is supported by X-ray diffraction analysis (XRD, Supplementary Fig. 3), where only diffraction peaks corresponding to $Li_2Si_2O_5$ for LDGC were observed before and after overcoating.

The consideration of aesthetics is crucial in the context of dental ceramics. Figure 1f shows that the veneer coated with $TiO_2/Al_2O_3$ (highlighted by a five-pointed star) exhibits no discernible difference compared to the pristine veneer (highlighted by a circle). Supplementary Fig. 4a shows the teeth that were not repaired with veneer. The alteration in color stability of the restoration affects the value and quality of the service to the patients[39]. Therefore, it is essential to evaluate whether there has been a noticeable change in color following ceramic modification. Experienced dentists first evaluated the visual inspection[40,41]. Supplementary Fig. 4b presents photos taken outside of the mouth, and visual inspection did not reveal any changes in gloss. This finding indicates its acceptability for wearers. Additionally, a colorimeter was employed to evaluate color changes of different samples, with $\Delta E$ values representing the deviation from the standard color. There were no significant alterations of $\Delta E$ values for LDGC, $Al_2O_3@LDGC$, $TiO_2@LDGC$, and $TiO_2/Al_2O_3@LDGC$ (Fig. 1g). Furthermore, Fig. 1h and Supplementary Fig. 4c demonstrated no difference in light irradiation intensity and light attenuation rate between LDGC and other samples. The exceptional aesthetic properties of ALD nanofilms are attributed to their uniform element distribution and optimal nanoscale thickness.

Raman spectroscopy revealed the chemical structure of ALD films on the surface of LDGC (Supplementary Fig. 5). The Si-O and Al-O bond peaks observed in LDGC remained unchanged after overcoating. No additional peak was detected in the Raman spectra of $Al_2O_3@LDGC$, while characteristic peaks of Ti-O, Ti-O-H, and Ti-O-Si were identified for $TiO_2@LDGC$ and $TiO_2/Al_2O_3@LDGC$. Therefore, chemical bonding between the ALD film and the LDGC substrate is established.

### Mechanical properties of the LDGC surface after overcoating

We employed a nano-scratch test to assess the mechanical properties of the surface after overcoating. The scratch position was determined based on the time-lateral displacement curve. For UP. CAD, in the case of $TiO_2$ overcoating, a critical point at 25.81 s indicated film detachment at this special point (Fig. 2a). Similarly, analysis of the time-normal force curve (Fig. 2b) revealed that $TiO_2$ nanolayer stripping

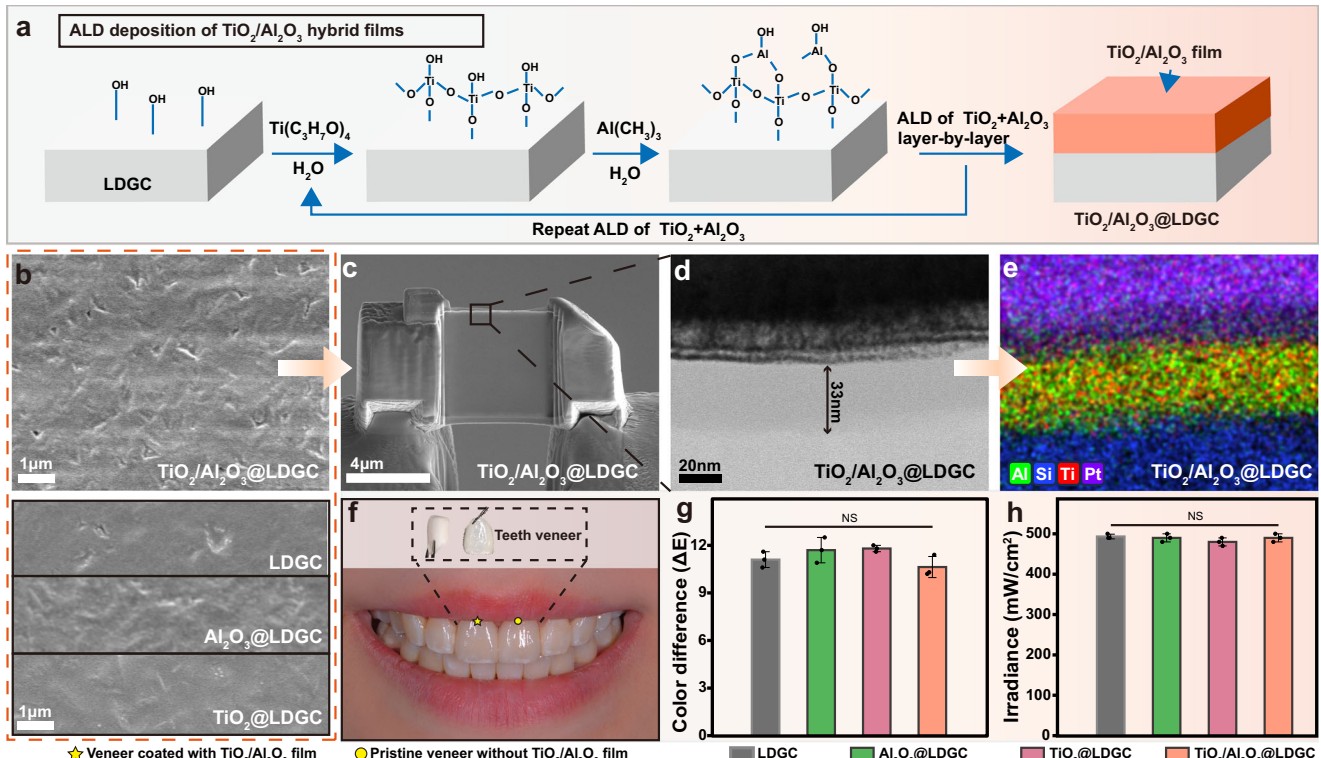

**Fig. 1 | Fabrication and characterization of surface film on LDGC. a** Schematic diagram of using ALD technology to prepare TiO$_2$/Al$_2$O$_3$ coated LDGC; **b** SEM for sample surface analysis of LDGC, Al$_2$O$_3$@LDGC, TiO$_2$@LDGC and TiO$_2$/Al$_2$O$_3$@LDGC; **c** FIB-SEM image of TiO$_2$/Al$_2$O$_3$@LDGC cross-section; **d** The film thickness was detected by transmission electron microscopy; **e** The element distribution of image d was analyzed by EDS; **f** Photos of veneer coated with TiO$_2$/ Al$_2$O$_3$ film and pristine veneer without TiO$_2$/Al$_2$O$_3$ film; **g** Color difference (ΔE) of LDGC and film coated LDGC samples; **h** Light irradiation intensity of LDGC and film coated LDGC samples. Statistical significance was analyzed using one-way ANOVA with Tukey's multiple comparisons test. NS is not significant. Data are presented as mean ± s.d. of $n$ = 3 biological replicates. Error bars represent s.d. Source data are provided as a Source Data file.

occurred when subjected to a normal force of 3202 μN at 25.81 s. The results of the nano-scratch test for LDGC without film detachment are in Supplementary Fig. 6. Moreover, the film detachment was also observed for LDGC coated with TiO$_2$-Al$_2$O$_3$ double nanolayer (TiO$_2$-Al$_2$O$_3$@LDGC and Al$_2$O$_3$-TiO$_2$@LDGC, Supplementary Fig. 7). In contrast, both Al$_2$O$_3$@LDGC (Fig. 2c, d) and TiO$_2$/Al$_2$O$_3$@LDGC (Fig. 2e, f) exhibited no signs of detachment even under a maximum loading force of up to 10000 μN. For IPS e.max CAD, in the case of TiO$_2$ overcoating, a critical point at 27.51 s indicated film detachment at this special point (Fig. 2g). Similarly, analysis of the time-normal force curve (Fig. 2h) revealed that TiO$_2$ nanolayer stripping occurred when subjected to a normal force of 4335 μN at 27.51 s. Similarly, Al$_2$O$_3$@LDGC (Fig. 2i, j) and TiO$_2$/Al$_2$O$_3$@LDGC (Fig. 2k, l) for IPS e.max CAD exhibited no signs of detachment, and the result was consistent with UP. CAD. This bonding ability can be attributed to the amorphous structure of Al$_2$O$_3$ and its improved interaction between the composite film and LDGC substrate[42]. Therefore, the presence of a TiO$_2$ nanolayer leads to easy peeling characteristics, which are inhibited when using the Al$_2$O$_3$ nanolayer or building the TiO$_2$/Al$_2$O$_3$ hybrid nanolayer.

Subsequently, the surface mechanical properties of the nanofilms were evaluated. The nanoindentation test revealed the surface hardness and reduced elastic modulus of the samples (Fig. 2m–p). The surface film had no significant effect on the reduced elastic modulus of LDGC. TiO$_2$/Al$_2$O$_3$@LDGC, Al$_2$O$_3$@LDGC, and TiO$_2$@LDGC exhibit superior hardness, highlighting its advantageous characteristics, while LDGC demonstrated comparatively lower values.

## Acid resistance properties of the coatings
ISO standards tested the chemical release of samples under conditions of 80 °C and 4% acetic acid. UP. CAD and IPS e.max CAD exhibited mass loss of 49.95 μg cm$^{-2}$ and 50.15 μg cm$^{-2}$, which meet ISO standards. SEM revealed that corrosion results in the formation of micropores on the surface of the LDGC (Supplementary Fig. 8). Dynamic corrosion experiments at 37 °C (with acetic acid solution replaced every 24 h) showed that after exposure to the acidic environment, the ceramic samples developed surface microporous structures at 24, 48, 72, and 96 h (Supplementary Fig. 8). ICP-MS analysis further indicated the continuous release of Li ions from the ceramics into the acidic solution, which is in line with previous findings (Supplementary Fig. 8)[26,43].

To reveal the acid resistance properties of the films, all samples (including UP. CAD and IPS e.max CAD) were immersed in artificial saliva at pH 6.8 and pH 2.5 to simulate neutral and acidic oral environments, respectively. For UP. CAD, SEM images revealed that after a 7-day immersion in neutral artificial saliva at pH 6.8, a small number of micropores existed on the surface of LDGC (Fig. 3a). When immersed in the environment at pH 2.5, the surface morphology of LDGC became rough and uneven, revealing weak acid resistance. Compared to the LDGC, both the Al$_2$O$_3$@LDGC surfaces before and after immersion in artificial saliva with a pH of 6.8 exhibited a smooth surface. However, numerous micropores similar to those in LDGC were formed, as observed for the Al$_2$O$_3$@LDGC group at pH 2.5, indicating the weak acid-resistant property of the Al$_2$O$_3$ film overcoating. In contrast, following immersion in acidic saliva at pH 2.5 for TiO$_2$@LDGC, higher surface smoothness was observed than that of the LDGC and Al$_2$O$_3$@LDGC, without any noticeable pore or rough surface structure. Similarly, TiO$_2$/Al$_2$O$_3$@LDGC showed no significant changes in surface morphology when exposed to an acidic liquid environment. For IPS e.max CAD, all samples displayed consistent results to UP. CAD (Fig. 3b). The SEM detection of hybrid TiO$_2$/Al$_2$O$_3$ coated and uncoated veneers was conducted in various pH environments, revealing that the film also effectively preserved the surface

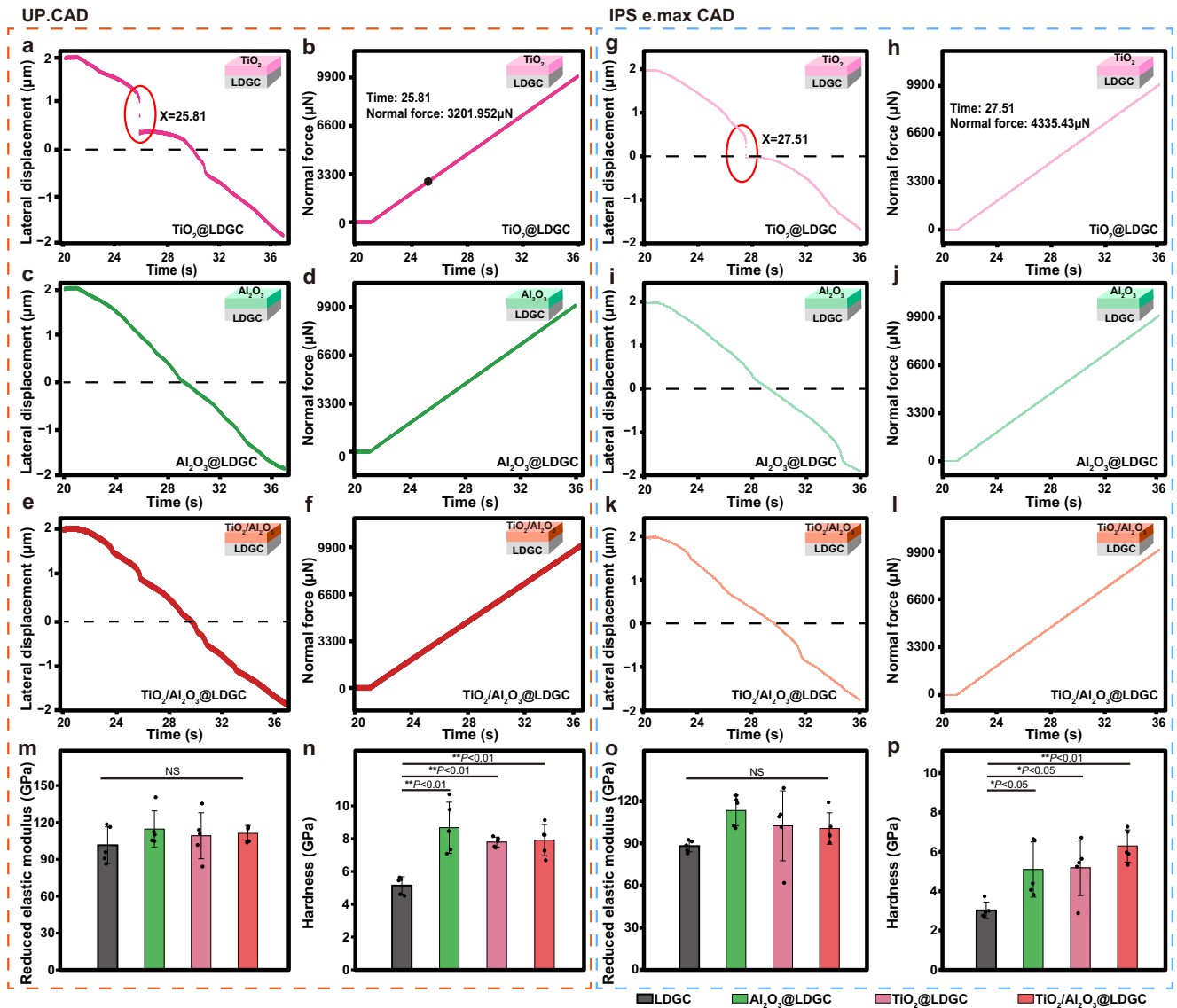

**Fig. 2 | Mechanical results of TiO₂ thin films, Al₂O₃ thin films, and TiO₂/Al₂O₃ thin films on LDGC.** Nano-scratch test of the time-lateral displacement data for **a**, **g** TiO₂@LDGC, **c**, **i** Al₂O₃@LDGC, **e**, **k** TiO₂/Al₂O₃@LDGC; Nano-scratch test of the time-normal force for **b**, **h** TiO₂@LDGC, **d**, **j** Al₂O₃@LDGC, **f**, **l** TiO₂/Al₂O₃@LDGC; The insets in (**a–l**) are schematic diagrams of LDGC coated with different thin films. The reduced elastic modulus (**m**, **o**) and hardness (**n**, **p**) of LDGC, TiO₂@LDGC, Al₂O₃@LDGC, TiO₂/Al₂O₃@LDGC. Statistical significance was analyzed using one-way ANOVA with Tukey's multiple comparisons test. For the reduced elastic modulus, NS is not significant. For the hardness of UP.CAD, $P$(LDGC-Al₂O₃@LDGC) < 0.01, $P$(LDGC-TiO₂@LDGC) < 0.01, $P$(LDGC-TiO₂/Al₂O₃@LDGC) < 0.01. For the hardness of IPS e.max CAD, $P$(LDGC-Al₂O₃@LDGC) < 0.05, $P$(LDGC-TiO₂@LDGC) < 0.05, $P$(LDGC-TiO₂/Al₂O₃@LDGC) < 0.01. Data are presented as mean ± s.d. of $n$ = 5 biological replicates. Error bars represent s.d. Source data are provided as a Source Data file.

microstructure of the veneers (Supplementary Fig. 9). Thus, as demonstrated in previous studies, the hybrid TiO₂/Al₂O₃ nanofilms deposited on complex surfaces possess acid resistance[44,45]. It can be concluded that TiO₂@LDGC and TiO₂/Al₂O₃@LDGC exhibited high acid resistance due to the ALD overcoating technology.

Surface roughness before and after acid immersion was assessed using a 3D measuring laser microscope, and the scanning range was 128 × 128 μm. The surface roughness of Al₂O₃@LDGC increased after acid immersion (pH 2.5), while TiO₂/Al₂O₃@LDGC remained at a similar level as the initial LDGC (Fig. 3c, d). These findings indicate that long-term exposure to an acidic oral environment can induce in vitro chemical corrosion of ceramics, consistent with previous research[46]. Notably, TiO₂/Al₂O₃ nanofilm effectively protected the glass ceramic surface from structural damage caused by acidic liquid.

The wear resistance of the samples immersed in both acidic and neutral environments was a crucial parameter for their clinical

application. Following immersion in an artificial saliva environment, surface wear was carried out using silicon nitride balls as grinding media, and the extent of surface wear was assessed using SEM. The image of the worn track displayed the depth and volume (Fig. 4a and d; the colour from red to black represents an increase in depth, with the unit of depth being μm). For UP. CAD, after being soaked in artificial saliva at a pH of 6.8, Al₂O₃@LDGC, TiO₂@LDGC, and TiO₂/Al₂O₃@LDGC exhibited relatively shallow surface wear marks compared to LDGC (Fig. 4a). Figure 4b shows the wear volume in a pH 6.8 environment. Al₂O₃@LDGC, TiO₂@LDGC, and TiO₂/Al₂O₃@LDGC exhibited a relatively low wear volume compared to LDGC. In contrast, when immersed in saliva at pH 2.5, all samples showed deeper wear marks than those at pH 6.8, and the deeper and wider marks observed on LDGC and Al₂O₃@LDGC surfaces were particularly noticeable. On the other hand, TiO₂@LDGC and TiO₂/Al₂O₃@LDGC exhibited relatively shallower wear marks overall. Figure 4c shows the wear volume in a pH 2.5 environment. The wear volume

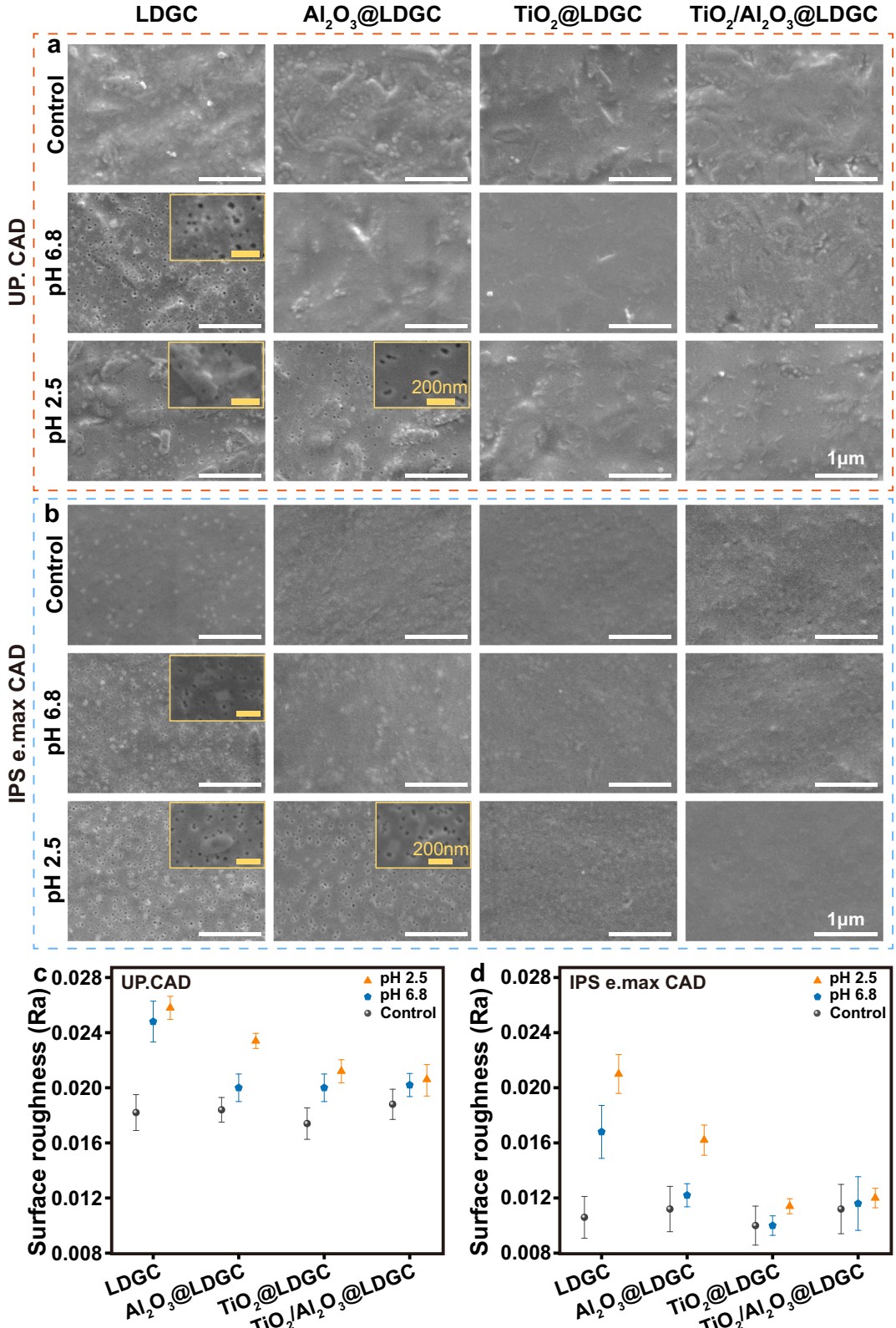

**Fig. 3 | Morphology and surface roughness of the LDGC after immersing in acidic artificial saliva.** SEM images of the LDGC surfaces of the four treatment groups in artificial saliva with different pH values **a** UP. CAD, **b** IPS e.max CAD, the insets of (**a**, **b**) are magnified SEM micrographs of the local area of the main panels. Surface roughness of different samples after solution immersion **c** UP. CAD, **d** IPS e.max CAD. Data are presented as mean ± s.d. of $n = 3$ biological replicates. Error bars represent s.d. Source data are provided as a Source Data file.

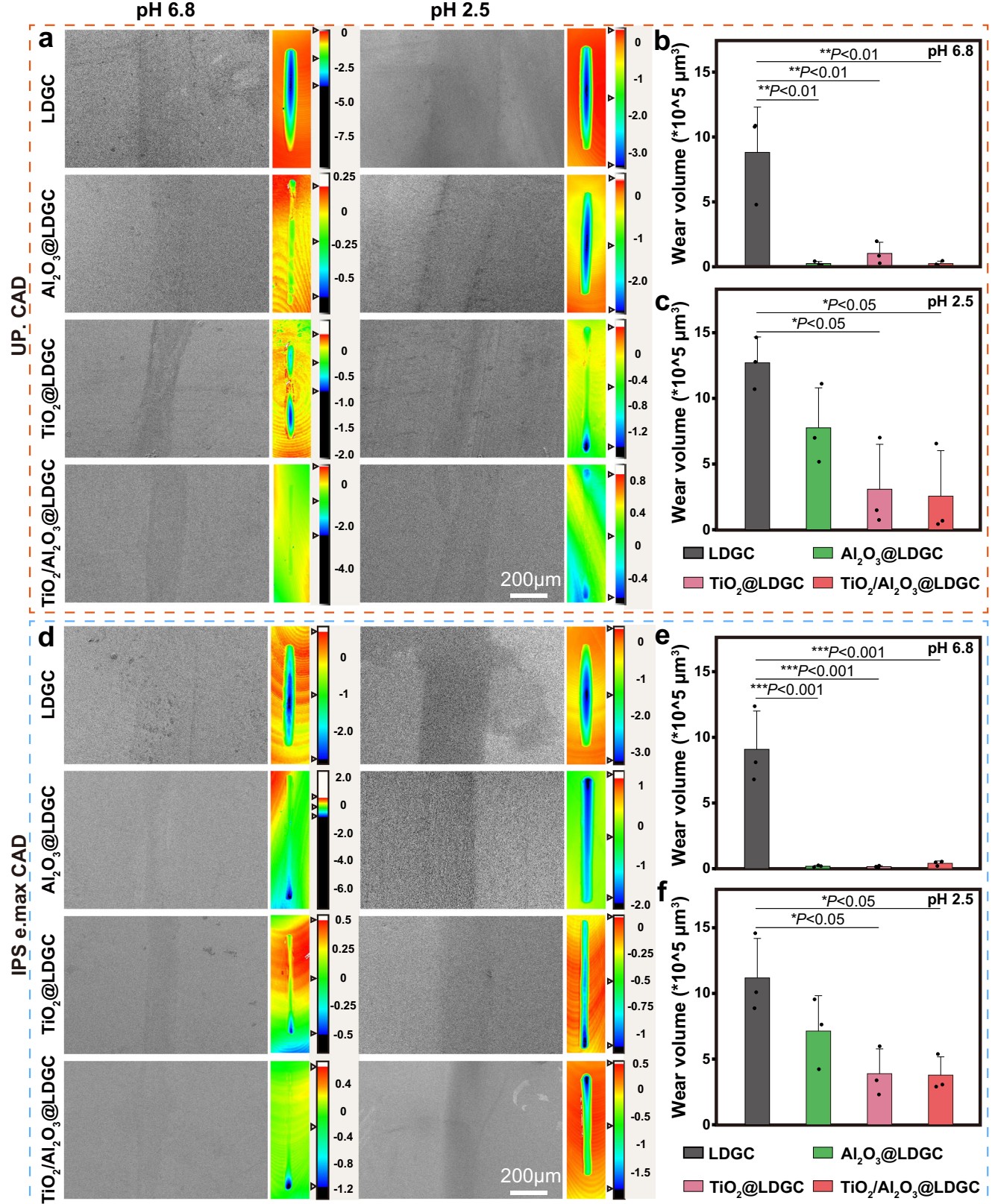

**Fig. 4 | Wear test of different specimens (LDGC, Al₂O₃@LDGC, TiO₂@LDGC, and TiO₂/Al₂O₃@LDGC) after immersion in artificial saliva with different pH values.** Images of SEM and the worn track of specimens (**a**) UP. CAD, (**d**) IPS e.max CAD; Wear volume of specimens immersed in saliva after pH 6.8 (**b**, **e**) and pH 2.5 (**c**, **f**). Statistical significance was analyzed using one-way ANOVA with Tukey's multiple comparisons test. For UP.CAD at the pH of 6.8, $P$(LDGC-Al₂O₃@LDGC) < 0.01, $P$(LDGC-TiO₂@LDGC) < 0.01, $P$(LDGC-TiO₂/Al₂O₃@LDGC) < 0.01. For UP.CAD at

the pH of 2.5, $P$(LDGC-TiO₂@LDGC) < 0.05, $P$(LDGC-TiO₂/Al₂O₃@LDGC) < 0.05. For IPS e.max CAD at the pH of 6.8, $P$(LDGC-Al₂O₃@LDGC) < 0.001, $P$(LDGC-TiO₂@LDGC) < 0.001, $P$(LDGC-TiO₂/Al₂O₃@LDGC) < 0.001. For IPS e.max CAD at the pH of 2.5, $P$(LDGC-TiO₂@LDGC) < 0.05, $P$(LDGC-TiO₂/Al₂O₃@LDGC) < 0.05. Data are presented as mean ± s.d. of $n$ = 3 biological replicates. Error bars represent s.d. Source data are provided as a Source Data file.

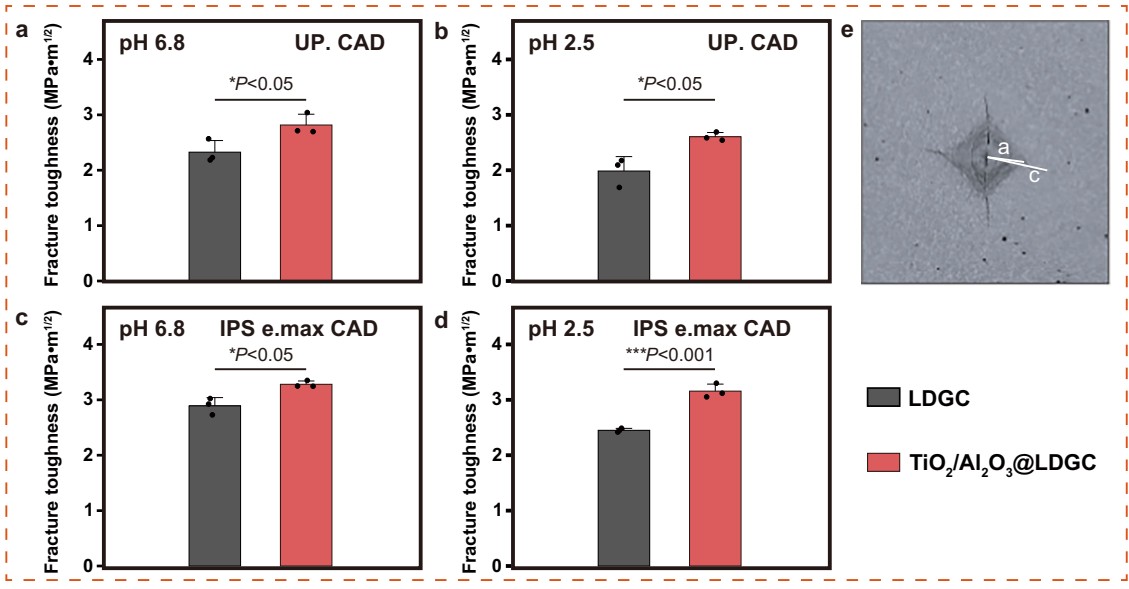

**Fig. 5 | The fracture toughness of the samples.** (**a**, **c**) pH 6.8; (**b**, **d**) pH 2.5; (**e**) Labeled fracture toughness test image (**c** is ½ of the mean length of the crack in μm, and (**a**) is ½ of the mean length between the diagonal lines of the indentation in μm). Statistical significance was analyzed using a *t*-test. For UP.CAD at the pH of 6.8, *P*(LDGC-TiO₂/Al₂O₃@LDGC) < 0.05. For UP.CAD at pH of 2.5, *P*(LDGC-TiO₂/ Al₂O₃@LDGC) < 0.05. For IPS e.max CAD at the pH of 6.8, *P*(LDGC-TiO₂/ Al₂O₃@LDGC) < 0.05. For IPS e.max CAD at the pH of 2.5, *P*(LDGC-TiO₂/ Al₂O₃@LDGC) < 0.001. Data are presented as mean ± s.d. of *n* = 3 biological replicates. Error bars represent s.d. Source data are provided as a Source Data file.

of Al₂O₃@LDGC increased compared to that in a pH 6.8. However, the wear volume of TiO₂@LDGC and TiO₂/Al₂O₃@LDGC still significantly decreased compared to LDGC. For IPS e.max CAD, the wear morphology is displayed in Fig. 4d, and the wear volume is displayed in Fig. 4e, f. Similarly, the wear volume of TiO₂@LDGC and TiO₂/Al₂O₃@LDGC was lower than that of LDGC.

To investigate the protective effect of thin films on the cracking properties of ceramics in an acidic environment, fracture toughness is measured to evaluate the efficacy of the films on the ceramic structure. The results showed that the LDGC after TiO₂/Al₂O₃ film coating exhibited higher fracture toughness (Fig. 5a–d), which may benefit the success rate of dental crown restoration. Figure 5e shows the labeled fracture toughness test image. This was obtained by pressing a Vickers indenter into the sample with a load of 0.1 N for 10 s. c is ½ of the mean length of the crack, and a is ½ of the mean length between the diagonal lines of the indentation. Among them, the fracture toughness of the sample was calculated based on the crack propagation.

**Elemental leaching and state of coatings after immersion**

In the acid resistance test, we found that TiO₂@LDGC and TiO₂/ Al₂O₃@LDGC exhibited exceptional morphological protection, wear resistance, and fracture resistance in an acid environment. To investigate this phenomenon further, we collected the immersion liquid after soaking the samples in the previous acid resistance experiment and employed inductively coupled plasma-mass spectrometry (ICP-MS) to analyze the ion release from each sample under both acidic and neutral environments. Figure 6a demonstrates a significant reduction in Li-ion release from both TiO₂@LDGC and TiO₂/Al₂O₃@LDGC compared to LDGC and Al₂O₃@LDGC at pH 2.5. In the immersion solutions of IPS e.max CAD (including TiO₂@LDGC and TiO₂/Al₂O₃@LDGC), the concentrations of Li and Si also decreased significantly under the pH 2.5 condition (Supplementary Fig. 10). Following immersion of ceramic samples in acidic environments, Li-ion release levels were significantly higher than those under neutral conditions. This is consistent with previous studies, which indicate that in an acidic environment, the destruction of the glass matrix structure is also facilitated by cation exchange between hydrated hydrogen ions and the ceramic surface, at

a faster rate than in a neutral environment[47]. The precipitation of soluble cations from the ceramic surface results in a porous hydrated silica layer[48]. Notably, the Ti concentration in the solution was significantly lower than that of Al even in the solution of pH 2.5 (Supplementary Figs. 10 and 11), indicating a higher acid resistance for TiO₂. This result aligns with Toma's previous research[49].

Raman spectra analysis was conducted for the sample after acid treatment. All samples exhibited broad bands at approximately ~359, ~413, ~468, and ~540–560 cm⁻¹, along with peaks at around ~934, ~950, and~1108 cm⁻¹. The peak at ~1108 cm⁻¹ can generally be attributed to the Li₂Si₂O₅ (Supplementary Fig. 12)[50,51], while Si-O stretching vibrations were observed at ~950 and ~934 cm⁻¹[52,53]. Peaks at ~920 cm⁻¹, ~788 cm⁻¹ were identified as the Ti-O bonds in TiO₂@LDGC and TiO₂/ Al₂O₃@LDGC, respectively[54,55]. The significant enhancement of the bond at ~933 cm⁻¹ may indicate the formation of Si-O-Ti bonds. No changes were observed in the Raman spectra for Al₂O₃@LDGC, possibly due to the presence of Al₂O₃ in the original LDGC (Supplementary Fig. 12). In contrast, TiO₂@LDGC and TiO₂/Al₂O₃@LDGC still displayed peaks of Ti-O bond at ~933, ~920, and ~788 cm⁻¹ when immersed in saliva with pH values of 6.8 and 2.5, indicating that TiO₂@LDGC and TiO₂/Al₂O₃@LDGC were stability in extreme acid environments (Fig. 6b and Supplementary Fig. 12).

We investigated the detailed changes in the electrical state of the elements on different films before and after acid treatments. The initial electric state of Ti 2*p* is the same on TiO₂/Al₂O₃@LDGC and TiO₂@LDGC (Fig. 6c). However, an increase in acidity of the immersion environments resulted in a shift towards higher binding energies for the Ti 2*p* peaks of TiO₂/Al₂O₃@LDGC. In contrast, no significant energy shift was observed for TiO₂@LDGC, indicating that the Ti plays a more crucial role in acid resistance. Furthermore, Fig. 6d shows that initial TiO₂/Al₂O₃@LDGC exhibited lower binding energies of Al 2*p* than Al₂O₃@LDGC due to the interaction between Ti and Al. The shift in Al binding energy and the decrease in intensity revealed corrosion of Al with increasing acidity in immersion environments. The lower binding energy of Al on TiO₂/Al₂O₃@LDGC compared to Al₂O₃@LDGC indicates that Al corrosion inhibition occurred in an acidic environment due to the chemical interaction between Ti and Al.

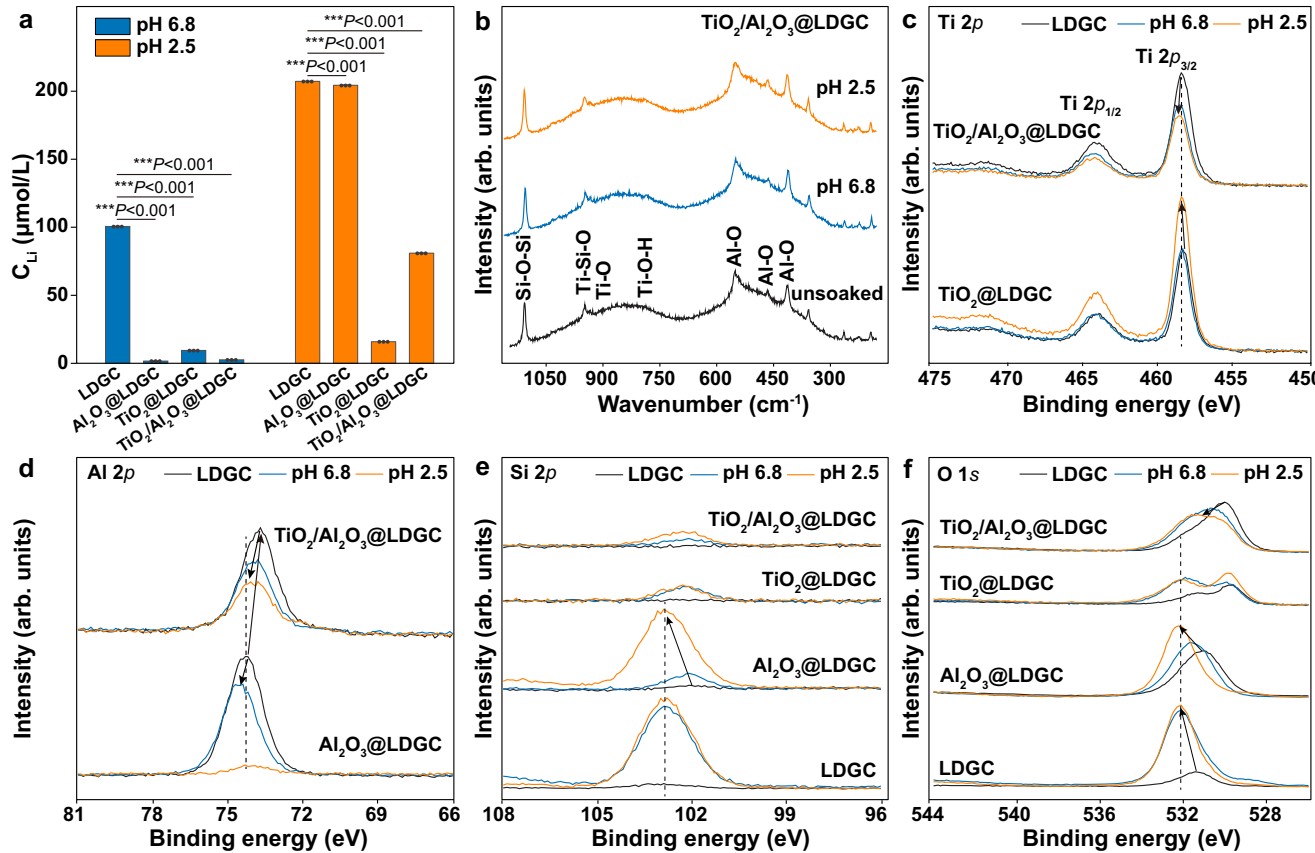

**Fig. 6 | Raman and XPS analysis of TiO₂/Al₂O₃@LDGC acid resistance mechanism. a** The concentration of released Li-ion in the artificial saliva. **b** Raman spectroscopy of samples for TiO₂/Al₂O₃@LDGC after immersion at different pH values. Electronic structure of **c** Ti 2*p*, **d** Al 2*p*, **e** Si 2*p*, **f** O 1*s* XPS spectroscopy of samples in these conditions that without immersion, after immersion in artificial saliva at pH 6.8, and after immersion in artificial saliva at pH 2.5, where arrows in (**c**–**f**) are used to indicate the obvious peak shifts (leftward or rightward) of the XPS

peaks among different samples. Statistical significance was analyzed using one-way ANOVA with Tukey's multiple comparisons test. For the pH 6.8, $P$(LDGC-Al₂O₃@LDGC) < 0.001, $P$(LDGC-TiO₂@LDGC) < 0.001, $P$(LDGC-TiO₂/Al₂O₃@LDGC) < 0.001. For the pH 2.5, $P$(LDGC-Al₂O₃@LDGC) < 0.001, $P$(LDGC-TiO₂@LDGC) < 0.001, $P$(LDGC-TiO₂/Al₂O₃@LDGC) < 0.001. Data are presented as mean ± s.d. of $n$ = 3 biological replicates. Error bars represent s.d. Source data are provided as a Source Data file.

The intensity of the Si 2*p* signal reflects the extent of LDGC exposure, and the binding energy revealed the interaction between ALD films and LDGC (Fig. 6e). The Si 2*p* signal intensity significantly decreased upon overcoating with 3 types of ALD films but increased after treatment in acid environments. Soaking Al₂O₃@LDGC in an acid solution with a pH of 2.5 enhanced the Si 2*p* signal intensity, comparable to that of LDGC, indicating the complete decomposition of the Al₂O₃ film. In contrast, TiO₂@LDGC and TiO₂/Al₂O₃@LDGC exhibited low Si 2*p* intensities, suggesting enhanced acid resistance. The acid dissolution and overcoating also changed the X-ray photoelectron spectra (XPS) of O 1*s* (Fig. 6f). The enhanced O 1*s* observed after acid treatment can be attributed to the dissolution of Li species and subsequent leaving of SiO₂ on the surface of LDGC and Al₂O₃@LDGC. In contrast, due to the overcoating, the characteristic O 1*s* peak originating from SiO₂ is obscured on TiO₂@LDGC and TiO₂/Al₂O₃@LDGC.

Although the Al element on TiO₂/Al₂O₃@LDGC is partially dissolved, the TiO₂/Al₂O₃@LDGC still showed high acid resistance in an acid environment. A cross-section of LDGC (Fig. 7a, b) and TiO₂/Al₂O₃@LDGC (Fig. 7d, e) were prepared after immersion in pH 2.5 saliva to elucidate the underlying mechanism. The element distribution in LDGC revealed the absence of Al and the presence of only Si elements, indicating complete dissolution of Al in the acid environment (Fig. 7c). In contrast, the surface layer of TiO₂/Al₂O₃@LDGC experienced a loss of Al element after being soaked in the acidic environment but remained at the contact interface of ALD film and substrate LDGC

(Fig. 7f). Additionally, the retention of Al within the LDGC matrix was still ensured through the protection of ALD composite film. This observation supports the formation of new chemical bonds between Al and Ti at the interface with Si, which may be one of the reasons why TiO₂/Al₂O₃@LDGC exhibits resistance to acid corrosion. Furthermore, after soaking in acidic saliva at pH 2.5, the Si/O ratio of TiO₂/Al₂O₃@LDGC was lower than that of LDGC (Supplementary Table 1), indicating reduced Si aggregation and reflecting the protective effect of the hybrid nanolayer on LDGC.

## Biosafety of surface coatings

The biosafety of surface film coating was subsequently evaluated. After culturing fibroblasts in an immersion solution of the samples for 48 h, qualitative cytotoxicity analysis was performed on the tested surfaces using a LIVE/DEAD™ Viability/Cytotoxicity Stain Kit. Live/dead fluorescent images revealed no significant difference in the number of dead cells compared to the control group (Supplementary Fig. 13a, b) and experimental groups. Specifically, the LDGC group (Supplementary Fig. 13c, d), Al₂O₃@LDGC (Supplementary Fig. 13e, f), TiO₂@LDGC (Supplementary Fig. 13g, h), and TiO₂/Al₂O₃@LDGC (Supplementary Fig. 13i, j) exhibited non-toxic properties towards cells. Cell viability was assessed using a CCK-8 reagent (Supplementary Fig. 13k). No significant difference in the cell viability was observed between the experimental and control groups' cells, indicating that LDGC, Al₂O₃@LDGC, TiO₂@LDGC, and TiO₂/Al₂O₃@LDGC were non-toxic to cells. Therefore, surface modification with the nanofilm on LDGC via

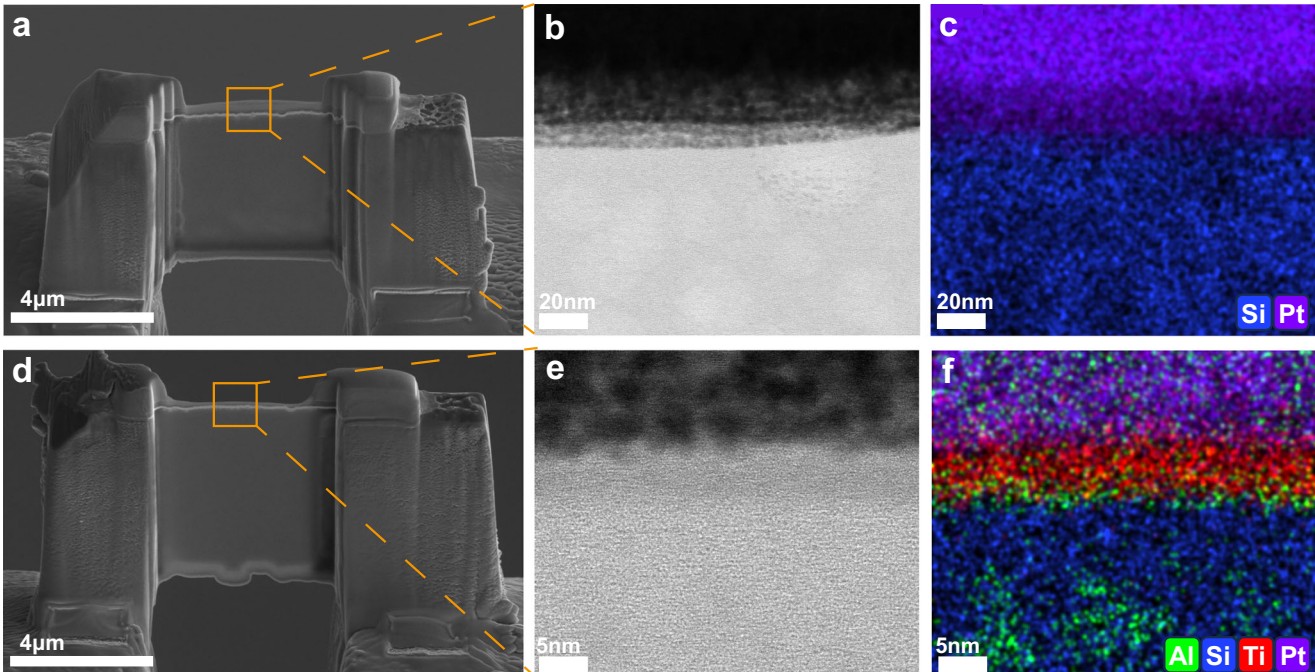

**Fig. 7 | Morphology and elemental distribution of the LDGC and TiO₂/Al₂O₃@LDGC cross-section after immersing in acidic artificial saliva.** **a** Morphology of LDGC cross-section prepared by FIB-SEM after immersion in pH 2.5 saliva; **b** STEM amplified the interface morphology of figure (**a**); **c** EDS showed the element distribution of figure (**b**); **d** The morphology of the TiO₂/Al₂O₃@LDGC cross-section prepared by FIB-SEM after immersion in pH 2.5 saliva; **e** STEM amplified the interface morphology of figure (**d**), (**f**) and EDS showed the element distribution of figure (**e**).

ALD is a simple, effective, and safe method to enhance surface properties.

## Discussion

Current research primarily focuses on enhancing the strength of dental ceramics, with limited attention to their protection in acidic oral environments. Our results show that even commercial crowns undergo acid corrosion (Supplementary Fig. 8), forming surface defects and pores that accelerate clinical failure and promote bacterial growth[24,26,43,56–58]. In this work, we employed ALD for surface coating of lithium silicate glass ceramics with TiO₂/Al₂O₃ hybrid nanofilms. This method forms a dense hybrid oxide nanofilm through chemical bonding. Compared to LDGC and Al₂O₃ nanolayer-coated LDGC, the overcoating of the TiO₂/Al₂O₃ hybrid nanolayer significantly enhances the acid resistance of LDGC, which effectively blocks acid solution corrosion, prevents the formation of micropores and cracks, and preserves the original wear resistance of the glass ceramics without affecting their light transmittance. Furthermore, comparing LDGC coated with the TiO₂ nanolayer and the TiO₂-Al₂O₃ double layer, the stripping of the nanolayer is completely suppressed in the case of LDGC coated with the TiO₂/Al₂O₃ hybrid nanolayer due to the atomic chemical bonding between Ti and Al oxidized species.

During practical operation, the prefabricated restorations were first adjusted to the desired occlusal height to ensure optimal comfort, followed by thin-film coating via ALD. Owing to the nanoscale thickness and high uniformity of the deposited film, no further adjustment of the coated ceramic is required. Additionally, owing to its precise control over composition, ALD technology holds promise for fabricating thin films with tunable mechanical properties and good biocompatibility, tailored to the specific mechanical requirements of different dental crowns. Moreover, the ALD process is automated and controlled via computer programs. Once the deposition protocol is established, the procedure merely requires selecting the pre-configured program on a computer to initiate and complete the coating process. Furthermore, the reaction chamber can accommodate multiple restorations simultaneously for thin-film deposition. Thus, these advantages, along with amenability to automation, make it easy for non-professionals to perform operations. The primary direction for the development of this technology involves advancing low-cost, dedicated ALD equipment and achieving clinical certification.

## Methods
### Materials
Artificial saliva (pH 6.8, including NaCl, KCl, CaCl₂, NaH₂PO₄, urea, Na₂S) and lactic acid (85%) were purchased from Shanghai Yuanye Bio-Technology Co., Ltd. Acetic acid (99.5%) was purchased from Shanghai Macklin Biochemical Co., Ltd. LDGC was provided by UPCERA (UP. CAD) and Ivoclar Vivadent AG (IPS e.max CAD). Dulbecco's Modified Eagle medium (DMEM, Gibco, composition of the DMEM in Supplementary Table 2) and fetal bovine serum (Gibco) were purchased from Thermo Fisher Scientific. The penicillin-streptomycin mixture was purchased from Beijing Solarbio Science & Technology Co., Ltd. LIVE/DEAD™ Viability/Cytotoxicity Stain Kit was purchased from Shanghai BestBio Biotechnology Co., Ltd. Cell Counting Kit-8 was purchased from Boster Biological Technology Co., Ltd. Titanium tetraisopropoxide (97%) was obtained from Alfa Aesar or Aldrich Chemical. Trimethyl aluminum (1.0 M solution in hexane) was obtained from Beijing Bailingwei Technology Co., Ltd.

### Specimens' preparation
Specimens made of LDGC were fabricated, and the size of the strip-shaped specimens was 20 × 4 × 1.6 mm. The size of sliced specimens was 13 × 13 × 0.5 mm to test the light irradiation intensity. Then, all specimens underwent sequential polishing with silicon carbide abrasive papers while being irrigated with water using a Precision Lapping Polishing Machine (UNIPOL-802, Shenyangkejing). The duration of use for #400-, #600-, #800-, #1200-, #1500-, and #2000-grit silicon carbide abrasive papers was approximately 1 - 2 h each. Finally, a mirror-

like polishing was achieved by applying a diamond grinding paste[12]. Additionally, dental veneers with a thickness of 0.5 mm were fabricated by computer-aided design and manufacturing (CAD/CAM, CEREC MC XL, Dentsply Sirona USA). All specimens and veneers were heat pressed following the manufacturer's guidelines in a heat press furnace (VACUMAT 6000 M, VITA). Finally, specimens and veneers were cleaned in ethanol and distilled water for 10 min using an ultrasonic cleaner, dried at 37 °C for use.

## Preparation of film by ALD

1). ALD of $TiO_2$. The $TiO_2$ ALD deposition was performed at 150 °C in the homemade ALD reactor, using titanium tetraisopropoxide and $H_2O$ as precursors. High-purity $N_2$ (99.999%) was utilized as a carrier gas. The heating temperature of titanium tetraisopropoxide is 80 °C, and $H_2O$ is room temperature. The samples were introduced into the ALD reactor, with the ALD process being controlled by a computer program. The ALD process involves pulsing and exposing the substrate to titanium tetraisopropoxide vapor, which initiates a half-reaction that consumes surface hydroxy groups. Subsequently, a purge process by $N_2$ flow was performed to eliminate unreacted titanium tetraisopropoxide and byproducts from the surface. The remaining ligands on the surface underwent further reaction with water, generating surface hydroxy groups for subsequent cycles via pulse, exposure, and purge of water vapor. These processes are referred to as 1 ALD cycle. The thickness of the $TiO_2$ is modulated by adjusting the number of ALD repeat cycles. The pulse, exposure, and purge times for the titanium tetraisopropoxide were 1, 8, and 30 s, and those for $H_2O$ were 0.1, 8, and 30 s, respectively.

2). ALD of $Al_2O_3$. The $Al_2O_3$ ALD deposition was carried out in the ALD reactor at 150 °C, by using trimethyl aluminum and $H_2O$ as precursors. The heating temperature of trimethyl aluminum and $H_2O$ is room temperature. The pulse, exposure, and purge times for trimethyl aluminum were 0.02, 8, and 35 s, respectively, and those for $H_2O$ were 0.1, 8, and 30 s, respectively.

3). Preparation of $TiO_2/Al_2O_3$@LDGC. The hybrid nanolayer of $TiO_2/Al_2O_3$ was deposited by integrating the ALD processes of $TiO_2$ and $Al_2O_3$ within a single cycle. One cycle of $TiO_2$ was followed by one cycle of $Al_2O_3$ to achieve a homogeneous mixture and interaction between $TiO_2$ and $Al_2O_3$ species at the atomic level. Through 150 repetitions of these combined cycles, a film composed of a $TiO_2$ and $Al_2O_3$ mixture was deposited onto LDGC, denoted as $TiO_2/Al_2O_3$@LDGC.

4). Preparation of $Al_2O_3$-$TiO_2$@LDGC. The double nanolayer of $Al_2O_3$-$TiO_2$ was deposited onto LDGC by sequentially depositing 150 cycles of $Al_2O_3$ followed by 150 cycles of $TiO_2$, denoted as $Al_2O_3$-$TiO_2$@LDGC. Only interface interactions occur between $Al_2O_3$ and $TiO_2$ nanolayers on $Al_2O_3$-$TiO_2$@LDGC.

5). Preparation of $TiO_2$-$Al_2O_3$@LDGC. The double nanolayer of $TiO_2$-$Al_2O_3$ was deposited onto LDGC by sequentially depositing 150 cycles of $TiO_2$ followed by 150 cycles of $Al_2O_3$, denoted as $TiO_2$-$Al_2O_3$@LDGC. Only interface interactions occur between $Al_2O_3$ and $TiO_2$ nanolayers on $TiO_2$-$Al_2O_3$@LDGC.

## Material characterization

The specimens were sputter-coated with thin Au films to provide a conductive surface for scanning electron microscopy. The surface of LDGC specimens was inspected by Field Emission Guns Scanning Electron Microscopy (FEGSEM, JSM-7900F, JEOL, Japan) at 2–5 kV. Elemental chemical analyses of the surfaces were performed using EDS (FlatQuad 5060 F, Bruker, Germany) coupled to the FEGSEM. X-ray diffraction (XRD, D8 ADVANCE, Bruker, Germany) was performed to investigate the crystalline structures of all specimens. Element concentrations in the samples were detected by ICP-MS (Agilent 7800,

America). The XPS spectra of the electronic state on the sample surface were recorded on a K-alpha photoelectron spectrometer (Thermo Scientific, USA). Raman spectroscopy was performed using WITec alpha300 RAS (Germany) with a 532 nm laser for detection. To characterize film thickness, samples were cut into cross-section using a FIB-SEM (Helios NanoLab 460HP, FEI, USA). The thickness of the thin film was characterized using a spherical-aberration-corrected transmission electron microscope (TEM, Titan Cubed Themis G2 300, FEI, USA), and the elemental distribution of the thin film was characterized using EDS (Super-X, Thermo Scientific, USA) coupled with TEM. The surface roughness of the samples was measured using a LEXT OLS5100 3D measuring laser microscope (Olympus, Japan), with a scanning range of $128 \times 128$ μm.

## The ΔE values of color changes

For strip-shaped specimens ($20 \times 4 \times 1.6$ mm), the surface color difference (ΔE) of the samples was observed using an advanced clinical portable dental spectrophotometer (VITA Easyshade® V, VITA Zahnfabrik, Germany), and the ΔE was recorded ($n = 3$, n is the sample number). Color comparison was carried out under indoor natural light according to the instrument instruction manual of the VITA Easyshade. Before each specimen measurement, VITA Easyshade was calibrated by placing a probe tip on the calibration port aperture. The measurement was considered valid when two consecutive, identical readings were generated for each area[59].

## Light irradiation intensity

For testing light irradiation intensity, sliced specimens ($13 \times 13 \times 0.5$ mm) were used to match the diameter of the light source. These results were performed by Cordless Light Cure (diameter = 10 mm, Bluephase N, Ivoclar Vivadent) and LED radiometer (Bluephase Meter II, Ivoclar Vivadent) ($n = 3$). The initial light intensity is 700 mW cm$^{-2}$.

$$\text{Light attenuation rate} = (1 - \text{Measuring Light Intensity}/\text{Intensity}) \times 100\%$$

(1)

## Nanoindentation tests and Nano-scratch tests

The strip-shaped specimens ($20 \times 4 \times 1.6$ mm) were used for tests. Nanoindentation tests (Hysitron TS 77 Select, Bruker, Germany) were performed to investigate any changes in the mechanical properties of the $TiO_2$@LDGC, $Al_2O_3$@LDGC, and $TiO_2/Al_2O_3$@LDGC specimens compared to LDGC specimens ($n = 5$). The loading method follows: loading force starts from 0 and increases to the maximum value of 200 μN at 5 s, keeps the load for 2 s, and then unloads. The testing device automatically determines the surface hardness and reduced elastic modulus from the unloading slope of the load-displacement curve and records both values.

Nano-scratch tests (Hysitron TS 77 Select, Bruker, Germany) were performed to investigate the interfacial adhesion strength of films on $TiO_2$@LDGC, $Al_2O_3$@LDGC, and $TiO_2/Al_2O_3$@LDGC ($n = 3$). Load-controlled nanoscratch tests were performed on the surface using the following load function: the normal load started at 0 mN and increased gradually to a peak of 10 mN over 4 μm of horizontal displacement.

## Immersion in the liquid environment

The acidity of artificial saliva was adjusted with lactic acid to prepare test solutions at pH 6.8 and pH 2.5. They were divided into four groups: LDGC, $TiO_2$@LDGC, $Al_2O_3$@LDGC, and $TiO_2/Al_2O_3$@LDGC. Each group of samples was divided into two equal parts. Artificial saliva at pH 6.8 and pH 2.5 was added to each part, respectively, and the samples were incubated in a 37 °C biochemical incubator for 7 days. Then,

the samples were taken out and washed twice with distilled water in an ultrasonic bath for 10 min each time. The liquid that had soaked the samples was collected, and the element concentrations were determined by ICP-MS.

## Linear reciprocating sliding wear tests

The samples were collected after the liquid immersion experiment for wear tests. The strip-shaped specimens ($20 \times 4 \times 1.6$ mm) were used. Reciprocating sliding tests ($n = 3$) were performed by a silicon nitride ball with 5 mm diameter at a rate of 120 mm min$^{-1}$, 0.8 N loading, and 3 mm displacement for 20 min. The reciprocating sliding tests were performed in a tribometer (UMT TriboLab, Bruker, Germany). Tests were carried out at 37 °C in artificial saliva. The morphology after wear tests was observed by FEGSEM. Wear track and volume were performed by a 3-dimensional confocal microscope (Phase Shift Micro-XAM-3D, USA).

## Fracture toughness

The samples were collected after the liquid immersion experiment for fracture toughness. The samples were tested for indentation on a Vickers microhardness tester with a load of 0.98 N for 10 s, which produced an acceptable crack pattern ($n = 3$). Mean crack length and diagonal length were obtained by a 3D measuring laser microscope. The fracture toughness ($K_{Ic}$) of the samples was estimated by the Eq. (2):

$$KIc = \frac{0.026\sqrt{E}\sqrt{P_a}}{\sqrt{c^3}}, \qquad (2)$$

where $K_{Ic}$ is the fracture toughness in MPa m$^{(1/2)}$, E is the Young's modulus in MPa, P is the load in N, c is ½ of the mean length of the crack in μm, and a is ½ of the mean length between the diagonal lines of the indentation in μm[12].

## Statistical analysis

All the data were analyzed using GraphPad Prism version 8.0. Results are reported as mean values and their standard deviations (s.d.). For datasets containing three or more groups, the statistical analysis involved a one-way analysis of variance (ANOVA) followed by Tukey's multiple comparisons test. When the dataset included two or fewer groups, a two-tailed $t$-test was applied to assess differences. Results are presented as mean values with corresponding standard deviations (s.d.). The $P$-value is defined as the probability of obtaining the observed data, or data more extreme, under the assumption that the null hypothesis is true. Statistical significance was considered for $P$-values less than 0.05 (NS is not significant, $*P < 0.05$, $**P < 0.01$, and $***P < 0.001$).

## Statistics and reproducibility

No statistical method was used to predetermine sample size. Sample sizes ($n = 3$–5) were selected based on common practice in the field and literature precedent. No data were excluded from the analyses. All replicates represent biological replicates. All samples and cells were allocated randomly into experimental groups. The investigators were not blinded to allocation during experiments and outcome assessment.

## Ethics statement

Intraoral photographs were obtained from participants. The images included both teeth without veneers and with veneers (with or without $TiO_2/Al_2O_3$ films), placed temporarily only for imaging purposes. These intraoral photographs do not contain names or any features that could identify individual patients. No bonding procedures or follow-up treatments were performed. Written informed consent was obtained

from each participant, and the study protocol was approved by the Medical Ethics Committee of Shanxi Medical University Stomatological Hospital (Approval No. 2024SLL002).

## Reporting summary

Further information on research design is available in the Nature Portfolio Reporting Summary linked to this article.

## Data availability

All the data generated in this study are provided in the Supplementary Information/Source Data file. Source data are provided with this paper.

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

## Acknowledgements

We acknowledge the financial support from the National Science Fund for Distinguished Young Scholars (Grant numbers 21825204 to Y.Q.), the National Natural Science Foundation of China (Grant numbers 22572209 and 22072172 to B.Z., 82071155 and 82271023 to X.W.), the Youth Innovation Promotion Association CAS (Grant numbers Y2021056 to B.Z.), the Basic Research Project of Shanxi Province (Grant numbers 202203021223006 to X.W.), and the Science and Technology Innovation Talent Team Project of Shanxi Province (Grant numbers 202204051001013 to Y.Q.). We acknowledge UPCERA for providing the substrate materials.

## Author contributions

B.Z., Y.Q., and X.W. directed the research. X.X.Z. performed the majority of the sample preparation, characterization, and data analysis. F.M., X.Y.Z., R.L., Y.Z., and H.L. contributed to the nanoindentation test. B.Z. and M.L. contributed to the XPS experiments. XX.Z., B.Z., Y.Q., and X.W. wrote the paper, with input from all of the other coauthors. F.W., J.B., Z.Y., and X.R. provided tooth veneers and helped with aesthetic data analysis. All the authors commented and reviewed the manuscript.

## Competing interests

The authors declare no competing interests. Patent applicant: Institute of Coal Chemistry, Chinese Academy of Sciences. Name of inventor(s): Bin Zhang, Xiaoxuan Zhang, Xing Wang, Xiaoyan Ren, Ruilin Liu, Zhuo Li. Application number: CN202510927849.1. Status of application: Pending. Specific aspect of manuscript covered in patent application: the thin film preparation method described in this manuscript is part of the content disclosed in the patent.
