## [Transparent Peer Review file · Nature Communications]

Hybrid TiO₂/Al₂O₃ nanolayer overcoating enhances dental lithium disilicate glass-ceramics acid resistance and surface mechanical properties

Corresponding Author: Professor Xing Wang

Version 0:

Reviewer comments:

Reviewer #1

(Remarks to the Author)

I would like to extend my gratitude to the authors for their commendable efforts in conducting this study. The research focuses on enhancing the acid resistance surface properties of lithium disilicate restorations by coating the ceramic surface with nanofilms using Atomic Layer Deposition (ALD). Lithium disilicate is widely favored by dentists for its superior esthetic qualities. Several studies in the literature have highlighted the detrimental effects of acidic environments on the wear and mechanical properties of lithium disilicate. Therefore, the findings of this study would undoubtedly be of interest to readers in the dental field. The work is original and supports the conclusions drawn by the authors. While the methodology is robust, it would benefit from the inclusion of additional details to facilitate reproducibility.

I would like to offer some minor comments to the authors:

General Comment: Proofreading is advisable as there are instances of word/sentence repetition.

1. Introduction:

The introduction is well-crafted, outlining the problem and providing an overview of previous attempts to enhance the acid resistance of ceramics.

- On page 3, the statement "Controlling strategic thickness on certain surfaces after glaze sintering poses challenges due to high-temperature sintering requirements and the inability to re-coat after tooth-wear grinding occurs" raises a question regarding whether this also applies to the deposited nanolayer coating.
- Similarly, on page 3, the authors assert, "Based on these findings, applying ALD technology to deposit dense nano oxide films onto LDGC crown surfaces would result in a highly stable and compact film due to the formation of strong chemical bonds." However, at this point, it remains an unconfirmed assumption. Please rephrase accordingly.

2. Results and Discussion + Supporting Information:

This section is well-presented, with results elucidated comprehensively and supported by informative figures. The methodology is detailed in the supplementary file "supporting information."

- Could you specify the number of specimens used?
- How was gloss alteration investigated, and who conducted the visual inspection?
- SEM image of the double nanolayer specimens is missing.
- Regarding specimen polishing, could you provide information on the duration of usage for each carbide abrasive paper?
- Page 3 in Supporting information file: Could you please explain how did you fulfill the sequential alternate ALD of TiO₂ and Al₂O₃ ? sounds similar to the double nanolayer. could you please make it more clear to the reader?
- Please explain the ALD process in more details. This help other researchers to reproduce the work.
- Could you please justify the used ALD parameters? Is the used process temprature standard ? Please add a reference.
- Were special lighting conditions employed during spectrophotometer examination of ΔE ?

These minor points, once addressed, would enhance the clarity and comprehensiveness of your manuscript.

Reviewer #2

(Remarks to the Author)

- While the manuscript offers a comprehensive and well-founded characterization of the innovative ALD method for nano-coating flat lithium disilicate glass ceramics, the title is misleading. It suggests a surface treatment for LDGC crowns, which are more complex samples not evaluated in the current manuscript.
- The clinical necessity driving the coating of LDGC with TiO₂/Al₂O₃ hybrid nanolayers is not thoroughly established. It appears to be more speculative than a genuine response to an identified clinical need.
- The following is more of a statement rather than a sentence for a manuscript: "The 35 global dental crown market valued at was \$1.8 billion in 2018, with projections to reach 36 \$2.9 billion by 2026.
- Inaccurate failure rate for LDGC from Abdulrahman et al., 2021
- Misspelling line 56, page 2: "creaking"
- "no visual inspection reveals any gloss alterations" is somewhat unclear. It could be rephrased for clarity, such as "Visual inspection did not detect any gloss alterations. The paragraph states that the visual inspection did not reveal gloss alterations for the materials mentioned, indicating that the nanofilm overcoating meets the requirements of dental ceramics. However, it does not explain what specific requirements are being met or why gloss alterations are significant in this context. Providing more context or background information would enhance understanding. Indeed, the paragraph mentions the results of the visual inspection but does not provide any further analysis or interpretation of these findings.
- LDGC micrographs depict some kind of debris on the polished surface (Figure 3A). Were the samples cleansed after polishing with carbide abrasive papers?
- Flexural strength testing - Although P represents the fracture load, referring to it as 'P-value' could be misleading as it might be interpreted as the probability value. Additionally, the small sample size (n=3) renders the results inconclusive.
- The colors green and red mentioned in Figure S1 are not discernible; instead, the figures appear in shades of gray.
- Color measurements, wear, and surface roughness analysis: further details are needed such as the sample size for each test, equipment used, background for color measurement, scan size area, etc.
- Analyzing volume loss would have been more effective in assessing wear resistance. The wear figures contradict the observed findings.
- The conclusion overstates the research findings: As shown in Figure 4R, the ALD-modified glass-ceramics did not inherently exhibit excellent strength. The results may have been overstated due to the insufficient power in the sample size. Additionally, Figure 4P does not demonstrate acid resistance but rather shows loss of integrity of the nanocoating.

Reviewer #3

(Remarks to the Author)

Dear Authors,

the paper that was submitted to Nature Communications addressed a issue related to chemical stability (acid resistance) and mechanical properties in the field of lithium disilicate glass-ceramics for dental application. The abstract and the introduction of your manuscript is not reviewing the state of the art deep enough.

Lithium disilicate glass-ceramics are well known in the dental field with outstanding chemical and mechanical properties which fulfill the Dental ISO Standard 6872:2019.

You never refer or even mentioned the main company who invented the lithium disilicate glass-ceramics at first. Two of the major products IPS e.max CAD and IPS e.max Press are available on the dental market for more than 15 years and fulfill the dental ISO Standard.

You have to refer this in your manuscript and explain, why their should be a problem mor detailed. Maybe you what to refer to lithium metasilicate glass-ceramics which might be not chemical stable or don't have the necessary mechanical properties.

The abstract also refers to glass-ceramics containing additional zirconia crystals (line 41-43 and Reference 7-11). This is not correct, because the companies marketing slogan "zirconia-reinforced lithium silicate glass ceramics" is scientifically incorrect. Please add a new reference "Lubauer et al (10.1016/j.dental.2021.12.013)". This paper showed that no crystalline zirconia appeared in this dental glass-ceramics.

Therefore, a major revision of your abstract, introduction and scope of the manuscript is necessary.

Version 1:

Reviewer comments:

Reviewer #3

(Remarks to the Author)

Dear authors,

thank you very much for you revised manuscript. Lithium disilicate glass-ceramics are widly used since more than 15 years. You already mentioned this fact. The dental companies developed and commercialized this type of glass-ceramics by fullfilling the dental ISO Standards.

There is no need to improve the chemical resistance, because their are no real issues on the market.

The statement, that chemical aeging would led to pores and cracks is quite difficult to explain or even true. May be you have to read the following text book and of course you have to put it in the reference list. Glass ceramic technology written by

Wolfram Höland and George Beall 2002 and the second edition 2012.

Kind regards

Reviewer #4

(Remarks to the Author)

The Authors addressed most of the Reviewer 2's comments and revised the paper properly. However, there are still a couple of issues deserving the Authors' attention:

1. With regard to the edited text "Specimens (substitutes for dental crowns) made of lithium disilicate glass- ceramic (LDGC) were fabricated...": according to the Reviewer comments on the crowns, are you sure that "substitutes for dental crown" is necessary? I suggest to delete it.
2. The statistical analysis is apparently missing. I guess that the Authors misunderstood the Reviewer 2's comment about P and P-value and deleted the definition of P-values. They could simply use a different symbol, but the statistical significance must be specified in the experimental section and, if appropriate, also in the legend of relevant figures; otherwise, interpretation and reliability of results become unclear and doubtful.

Version 2:

Reviewer comments:

Reviewer #3

(Remarks to the Author)

Dear Authors,

thank you very much for submitting your answers, interpretation and revised manuscript.
Of course, you presented a lot of interesting results and a interesting method within your paper.
But, still I don't belief that it is neccessary to coat the surface of a dental restoration with a special nano-layer.

It is not discussed, that the effort to coat one or more restorations is really high and need educated persons. On the other hand you have to realize that after the process of fixation into the patients mooth the endodontics will be checked. That means, the dentist will remove some how the surface of a crown, inlay, onlay etc. Do you belive that the coated layer will be still there?

Please, you have to discuss this topics or make some remarks on that.

Best wishes

Reviewer #4

(Remarks to the Author)

The Reviewer's comments were addressed properly. The revision is adequate and I have no further comment to add; the ms can be accepted.

Below please find our responses to the comments and questions from the reviewers:

Reviewer #3 (Remarks to the Author):

Dear authors, thank you very much for you revised manuscript. Lithium disilicate glass-ceramics are widely used since more than 15 years. You already mentioned this fact. The dental companies developed and commercialized this type of glass-ceramics by fulfilling the dental ISO Standards.

1. There is no need to improve the chemical resistance, because their are no real issues on the market.

Response:

Thanks for your suggestion. We agree that dental companies have developed and commercialized glass ceramics by fully complying with the dental ISO standards. According to the latest ISO standard (2024), the permissible weight change limit is less than $100 \mu\text{g}/\text{cm}^2$ by soaking the specimens in 4% acetic acid at $80 \text{ }^\circ\text{C}$ for 16 hours. We conducted the testing experiments according to ISO standards. The UP.CAD and IPS e.max CAD specimens exhibit mass loss of $50.15 \mu\text{g}/\text{cm}^2$ and $49.96 \mu\text{g}/\text{cm}^2$, respectively. Both values fully meet the ISO standards.

However, multiple clinical studies have shown that ceramic materials that meet the ISO standards still suffer from cracking during clinical applications.^[1, 2] The dental implant crowns, molars, and other heavy occlusion areas face a further increased risk of fractures, especially during nocturnal bruxism and gastric reflux.^[3-6] Patients with nocturnal bruxism and gastric reflux have to choose harder zirconia materials but in the loss of aesthetic effects. Zirconia can be doped into lithium silicate ceramics to increase hardness, but it will reduce the translucency of dental crowns and increase the difficulty in the cutting process.

Notably, the effect of chemical corrosion on the surface structure and morphology of the ceramic has received limited attention.^[7] It has been reported that acid-base environments lead to the formation of micropores on the surface.^[8, 9] Moreover, increased corrosion time causes a gradual increase in ion release, potentially allowing corrosive media to penetrate deeper into the material structure.^[10, 11] These surface micropores and subsurface erosion provide favorable conditions for cracking initiation. Dynamic pH corrosion testing further indicates that corrosion reduces flexural

strength,^[6] resulting in an overall strength decline in glass-ceramics. Under acidic conditions, microcracks can form on the surface of glass ceramics, analogous to the scoring created by a glass knife on glass, which significantly increases the likelihood of fracture. Therefore, preventing glass ceramic failure involves enhancing their intrinsic mechanical strength and minimizing the formation of surface scratches or flaws that act as initiation sites for crack propagation.

As noted by the reviewers, our study strictly followed the experimental methodology outlined in ISO 6872. Our results further demonstrated that the test materials meet the allowable mass change limit of less than 100 $\mu\text{g}/\text{cm}^2$. However, the applied glass-ceramics indeed suffer from corrosion in liquid media, particularly under highly acidic conditions which increases ion release and damages the surface microstructure. We have also conducted the mild corrosion control experiment at a simulated body temperature (37 °C instead of 80 °C). The simulations followed the ISO-specified parameters: samples with a surface area exceeding 30 cm^2 were immersed in a 4% acetic acid solution. The immersion medium was replaced every 24 hours. Immersion solutions were collected for Li-ion release analysis at 24, 48, 72, and 96 hours. The results indicated continuous release of Li-ion (Supplementary Fig. 8). Scanning electron microscopy (SEM) revealed that the corrosion occurred and led to numerous microporous structures on the surface (Supplementary Fig. 8).

In short, evidence from the literature, clinical observations, and our current studies demonstrates that ceramics meeting international standards still encounter challenges in acidic environments. However, such degradation in performance has not received enough attention. This underscores the need for continuous research in this field. Different from all previous methods by adding fillers, our work introduces an innovative approach: coating chemically bonded thin films on ceramic surfaces, which effectively prevents direct contact between acidic solutions and ceramic constituents, thereby inhibiting corrosion at the interface. Notably, these nanoscale films—undetectable during mastication and exerting no impact on ceramic color—uniformly cover surface grooves of ceramic crowns through robust chemical bonding, ensuring complete isolation of the substrate from corrosive environments. By establishing this novel surface modification mechanism, our work opens new frontiers for ceramic surface functionalization and triggers a chain of follow-up investigations across more

research groups.

We want to express our gratitude to the reviewers for their valuable comments, including the previously mentioned considerations of a broader range of ceramic materials and the reasons for the research on enhanced strength. The above-mentioned results and discussions have been added in the revised manuscript.

References:

1. Hjerpe J, et al. Minimally invasive glass-ceramic restorations: Clinical and patient-reported outcomes in full-mouth rehabilitations. *J Dent* **154**, 105571 (2025).
2. Aziz AM, El-Mowafy O, Tenenbaum HC, Lawrence HP. Clinical performance of CAD-CAM crowns provided by predoctoral students at the University of Toronto. *J Prosthet Dent* **127**, 729-736 (2022).
3. Çavusoglu Y, Sahin E, Gürbüz R, Akça K. Fatigue Resistance of 2 Different CAD/CAM Glass-Ceramic Materials Used for Single-Tooth Implant Crowns. *Implant Dent* **20**, 374-378 (2011).
4. Beier US, Kapferer I, Burtscher D, Dumfahrt H. Clinical performance of porcelain laminate veneers for up to 20 years. *Int J Prosthodont* **25**, 79-85 (2012).
5. Jiang Q, Wang Z, Zhang S, Liu X, Fu B. Performance of Bonded Lithium Disilicate Partial-coverage Crowns in the Restoration of Endodontically Treated Posterior Teeth: An Up to Seven-Year Retrospective Study. *Oper Dent* **49**, 365-375 (2024).
6. Esquivel-Upshaw JF, et al. Fracture of Lithia Disilicate Ceramics under Different Environmental Conditions. *Materials* **15**, 5261 (2022).
7. Fathy SM, Swain MV. In-vitro wear of natural tooth surface opposed with zirconia reinforced lithium silicate glass ceramic after accelerated ageing. *Dent Mater* **34**, 551-559 (2018).
8. Milleding P, Wennerberg A, Alaeddin S, Karlsson S, Simon E. Surface corrosion of dental ceramics in vitro. *Biomaterials* **20**, 733-746 (1999).
9. Esquivel-Upshaw JF, Dieng FY, Clark AE, Neal D, Anusavice KJ. Surface degradation of dental ceramics as a function of environmental pH. *J Dent Res* **92**, 467-471 (2013).
10. Švančárková A, Galusková D, Nowicka AE, Pálková H, Galusek D. Effect of corrosive media on the chemical and mechanical resistance of IPS e.max® CAD based Li₂Si₂O₅ glass-ceramics. *Materials* **15**, 365 (2022).
11. Hsu SM, et al. Effect of pH cycling frequency on glass-ceramic corrosion. *Materials* **13**, 3655 (2020).

Changes:

- 1) "Exposure to acidic conditions in the mouth will lead to the formation of pores and crack." was modified as "Exposure LDGC to acidic conditions within the oral cavity leads to pore formation, and crack risk increases after additional long-term wear." (Page 1, lines 22)
- 2) "ISO standards tested the chemical release of samples under conditions of 80°C and 4% acetic acid. UP. CAD and IPS e.max CAD exhibited mass loss of 49.95 µg/cm² and 50.15 µg/cm², which meet ISO standards. SEM revealed that corrosion results in the formation of micropores on the surface of the LDGC (Supplementary Fig. 8). Dynamic corrosion experiments at 37°C (with acetic acid solution replaced every 24 hours) showed that after exposure to the acidic environment, the ceramic samples developed

surface microporous structures at 24, 48, 72, and 96 hours (Supplementary Fig. 8). ICP-MS analysis further indicated the continuous release of Li ions from the ceramics into the acidic solution, which is in line with previous findings (Supplementary Fig. 8).^[22, 39] was added in the revised manuscript. (Page 10, lines 3-13)

Supplementary Figure 8 | Corrosion tests were conducted by immersing samples in a 4% acetic acid solution. SEM of LDGC soaked in 4% acetic acid (a) UP.CAD, (c) IPS e.max CAD; Li-ion release from LDGC by ICP-MS (b) UP.CAD, (d) IPS e.max CAD. Statistical significance was analyzed using one-way ANOVA with Tukey's multiple comparisons test (n = 3). $P(80\text{ °C}-37\text{ °C}) < 0.001$. Data are presented as mean \pm s.d. Source data are provided as a Source Data file.

2.The statement, that chemical ageing would led to pores and cracks is quite difficult to explain or even true. May be you have to read the following text book and of course you have to put it in the reference list. Glass ceramic technology written by Wolfram Höland and George Beall 2002 and the second edition 2012.

Response: Thank you for your suggestion. We have carefully read the book you recommended and gained a deeper understanding of lithium silicate ceramics. And we have cited this book in the introduction (reference 23).

The text highlights that lithium silicate ceramics have crystalline and glassy phases. When characterizing the internal crystalline architecture of glass-ceramics, acid etching of the amorphous matrix results in a porous microstructure induced by corrosion. The enhancement of chemical durability in glass-ceramics is attributed to two key factors: the inclusion of nucleating agents—such as phosphorus pentoxide (P₂O₅) and metal-based nucleators—and the incorporation of metal oxide constituents like aluminum oxide (Al₂O₃) and potassium oxide (K₂O) in the glass composition. However, their total

concentration must be constrained within a specific range to maintain structural integrity. While glass-ceramics with complex crystalline structures exhibit superior chemical durability compared to their parent glass, certain soluble constituents in the glassy matrix can still leach out, leading to the formation of micropores.

This book clearly demonstrates the problem of surface micropore formation in glass ceramics during use. The closely related contents in the book (*Glass-Ceramic Technology*) are as follows.

1) **Page 1:** "A glass-ceramic may be highly crystalline or may contain substantial residual glass. It is composed of one or more glassy and crystalline phases."

2) **Page 250:** "The chemical durability of glass-ceramics can vary over a wide range depending upon bulk chemistry, the durability of each of the crystalline phases, and the microstructure. Typically, in highly crystalline silicate glass ceramics, the durability of the glass - ceramic may be somewhat superior to that of the original glass. There are exceptions to this rule, however, particularly when a soluble component is concentrated in a continuous residual glass. Figure 3.31 a, b is a transmission electron microphoto of a β -spodumene glass-ceramic with 5 wt% B_2O_3 added. This component can be seen to concentrate in the siliceous residual glass and to spontaneously separate into a boron-rich continuous phase. This phase is soluble in aqueous solutions, and because it forms a continuous film around the durable β -spodumene crystals, this glass-ceramic had very poor chemical resistance in comparison with nonboron compositions and even compared with its parent glass."

Figure 3.31b. Higher magnification illustrating boron-rich (white) continuous glassy portion of residual glass accounting for poor chemical durability of the glass-ceramic.

3) **Page 78:** "Lithium metasilicate crystals are characterized by the ease with which

they dissolve from the glass-ceramic in dilute hydrofluoric acid."

In addition to the content in the book, more recent researches have also demonstrated the importance of chemical resistance, such as the absence of amorphous phase confinement renders the material susceptible to brittle fracture¹². That corrosion by acidic solutions decreases the flexural strength of ceramics⁶. The related effects on chemical durability, such as the release of Li-ion and the time for surface structure corrosion, are still unclear. We listed references 7-11 in **the previous response**, focusing on the changes in surface structure and mechanical properties caused by chemical corrosion. Our supplementary data showed that immersion in artificial saliva and acetic acid solutions induced surface corrosion, leading to structural modifications. Exposure to the ISO standard test solution revealed detectable surface structural changes within 24 hours, with progressive alterations observed over time (Supplementary Fig. 8). We believe that ongoing research will drive the advancement of lithium disilicate ceramics, enabling them to withstand diverse in vivo environmental variations and better serve patient needs.

In summary, in contrast to conventional strategies that rely on adding filler, our work proposes an innovative approach: coating chemically bonded thin films on ceramic surfaces by atomic layer deposition. This conformal coating precisely replicates/maintains the topography of restoration at the atomic level. While maintaining original occlusal functionality and aesthetic integrity, the coated films effectively prevent direct contact between acidic solutions and ceramic constituents, thereby inhibiting corrosion at the interface.

References:

12. Zhao H, et al. Multiscale engineered artificial tooth enamel. *Science* **375**, 551-556 (2022).
6. Esquivel-Upshaw JF, et al. Fracture of Lithia Disilicate Ceramics under Different Environmental Conditions. *Materials* **15**, 5261 (2022).

Changes:

- 1) We have incorporated reference 23 into the introduction section.

Reviewer #4 (Remarks to the Author):

The Authors addressed most of the Reviewer 2's comments and revised the paper properly. However, there are still a couple of issues deserving the Authors' attention:

1. With regard to the edited text “Specimens (substitutes for dental crowns) made of lithium disilicate glass-ceramic (LDGC) were fabricated...”: according to the Reviewer comments on the crowns, are you sure that “substitutes for dental crown” is necessary? I suggest to delete it.

Response: Thank you for your suggestion. Following your suggestion, we have deleted "substitutes for dental crown".

Changes: "substitutes for dental crown" was deleted. (Page 3, line 20, Supplementary information)

2. The statistical analysis is apparently missing. I guess that the Authors misunderstood the Reviewer 2’s comment about P and P-value and deleted the definition of P-values. They could simply use a different symbol, but the statistical significance must be specified in the experimental section and, if appropriate, also in the legend of relevant figures; otherwise, interpretation and reliability of results become unclear and doubtful.

Response: We greatly appreciate your comments. We have added the significance of statistical methods and the definition of P-values in the figure legends and methods to make the manuscript more comprehensive.

Changes:

1) "Statistical significance was analyzed using one-way ANOVA with Tukey’s multiple comparisons test ($n = 3$). NS is not significant. Data are presented as mean \pm s.d. Source data are provided as a Source Data file." was added in the revised manuscript. (Page 6, Fig. 1)

2) "Statistical significance was analyzed using one-way ANOVA with Tukey’s multiple comparisons test ($n = 5$). For the reduced elastic modulus, NS is not significant. For the hardness of UP.CAD, $P(\text{LDGC-Al}_2\text{O}_3@LDGC) < 0.01$, $P(\text{LDGC-TiO}_2@LDGC) < 0.01$, $P(\text{LDGC-TiO}_2/\text{Al}_2\text{O}_3@LDGC) < 0.01$. For the hardness of IPS e.max CAD, $P(\text{LDGC-Al}_2\text{O}_3@LDGC) < 0.05$, $P(\text{LDGC-TiO}_2@LDGC) < 0.05$, $P(\text{LDGC-TiO}_2/\text{Al}_2\text{O}_3@LDGC) < 0.01$. Data are presented as mean \pm s.d. Source data are provided as a Source Data file." was added in the revised manuscript. (Page 9, Fig. 2)

3) "($n = 3$ independent samples per group). Data are presented as mean \pm s.d. Source data are provided as a Source Data file." was added in the revised manuscript. (Page 13, Fig. 3)

4) "Statistical significance was analyzed using one-way ANOVA with Tukey's multiple comparisons test ($n = 3$). For UP.CAD at the pH of 6.8, $P(\text{LDGC-Al}_2\text{O}_3@LDGC) < 0.01$, $P(\text{LDGC-TiO}_2@LDGC) < 0.01$, $P(\text{LDGC-TiO}_2/\text{Al}_2\text{O}_3@LDGC) < 0.01$. For UP.CAD at the of pH 2.5, $P(\text{LDGC-TiO}_2@LDGC) < 0.05$, $P(\text{LDGC-TiO}_2/\text{Al}_2\text{O}_3@LDGC) < 0.05$. For IPS e.max CAD at the pH of 6.8, $P(\text{LDGC-Al}_2\text{O}_3@LDGC) < 0.001$, $P(\text{LDGC-TiO}_2@LDGC) < 0.001$, $P(\text{LDGC-TiO}_2/\text{Al}_2\text{O}_3@LDGC) < 0.001$. For IPS e.max CAD at the pH of 2.5, $P(\text{LDGC-TiO}_2@LDGC) < 0.05$, $P(\text{LDGC-TiO}_2/\text{Al}_2\text{O}_3@LDGC) < 0.05$. Data are presented as mean \pm s.d. Source data are provided as a Source Data file." was added in the revised manuscript. (Page 14-15, Fig. 4)

5) " Statistical significance was analyzed using a t-test ($n = 3$). For UP.CAD at the pH of 6.8, $P(\text{LDGC-TiO}_2/\text{Al}_2\text{O}_3@LDGC) < 0.05$. For UP.CAD at pH of 2.5, $P(\text{LDGC-TiO}_2/\text{Al}_2\text{O}_3@LDGC) < 0.05$. For IPS e.max CAD at the pH of 6.8, $P(\text{LDGC-TiO}_2/\text{Al}_2\text{O}_3@LDGC) < 0.05$. For IPS e.max CAD at the pH of 2.5, $P(\text{LDGC-TiO}_2/\text{Al}_2\text{O}_3@LDGC) < 0.001$. Data are presented as mean \pm s.d. Source data are provided as a Source Data file." was added in the revised manuscript. (Page 15-16, Fig. 5)

6) "Statistical significance was analyzed using one-way ANOVA with Tukey's multiple comparisons test ($n = 3$). For the pH 6.8, $P(\text{LDGC-Al}_2\text{O}_3@LDGC) < 0.001$, $P(\text{LDGC-TiO}_2@LDGC) < 0.001$, $P(\text{LDGC-TiO}_2/\text{Al}_2\text{O}_3@LDGC) < 0.001$. For the pH 2.5, $P(\text{LDGC-Al}_2\text{O}_3@LDGC) < 0.001$, $P(\text{LDGC-TiO}_2@LDGC) < 0.001$, $P(\text{LDGC-TiO}_2/\text{Al}_2\text{O}_3@LDGC) < 0.001$. Data are presented as mean \pm s.d. Source data are provided as a Source Data file." was added in the revised manuscript. (Page 18, Fig. 6)

7) "Statistical analysis

All the data were analyzed using GraphPad Prism version 8.0. Results are reported as mean values and their standard deviations (s.d.). For datasets containing three or more groups, statistical analysis involved one-way analysis of variance (ANOVA) followed by Tukey's multiple comparisons test. When the dataset included two or fewer groups, a t-test was applied to assess differences. Results are presented as mean values with corresponding standard deviations (s.d.). The P-value is defined as the probability of obtaining the observed data, or data more extreme, under the assumption that the null hypothesis is true. Statistical significance was considered for P -values less than 0.05 (NS, not significant, $*P < 0.05$, $**P < 0.01$, and $***P < 0.001$)." was added in the revised manuscript. (Page 11, Supplementary information)

8) "Supplementary Figure 10 | Concentration of the element released in solution (UP. CAD). (a) Ti element. Statistical significance was analyzed using a t-test (n = 3). NS is not significant. For pH 2.5, $P(\text{TiO}_2@LDGC\text{-TiO}_2/\text{Al}_2\text{O}_3@LDGC)<0.001$; (b) Al element. Statistical significance was analyzed using a t-test (n = 3). For pH 6.8, $P(\text{Al}_2\text{O}_3@LDGC\text{-TiO}_2/\text{Al}_2\text{O}_3@LDGC)<0.01$; For pH 2.5, $P(\text{Al}_2\text{O}_3@LDGC\text{-TiO}_2/\text{Al}_2\text{O}_3@LDGC)<0.001$; Data are presented as mean \pm s.d. Source data are provided as a Source Data file." was added in the revised manuscript. (Page 21, Figure S10, Supplementary information)

9) "Supplementary Figure 11 | Concentration of the element released in solution (IPS e.max CAD). (a) Li element. Statistical significance was analyzed using one-way ANOVA with Tukey's multiple comparisons test (n = 3). $P(LDGC\text{-Al}_2\text{O}_3@LDGC)<0.001$, $P(LDGC\text{-TiO}_2@LDGC)<0.001$, $P(LDGC\text{-TiO}_2/\text{Al}_2\text{O}_3@LDGC)<0.001$; (b) Si element. Statistical significance was analyzed using one-way ANOVA with Tukey's multiple comparisons test (n = 3). For pH 6.8, $P(LDGC\text{-Al}_2\text{O}_3@LDGC)<0.001$, $P(LDGC\text{-TiO}_2@LDGC)<0.001$, $P(LDGC\text{-TiO}_2/\text{Al}_2\text{O}_3@LDGC)<0.001$; For pH 2.5, $P(LDGC\text{-Al}_2\text{O}_3@LDGC)<0.001$, $P(LDGC\text{-TiO}_2@LDGC)<0.05$, $P(LDGC\text{-TiO}_2/\text{Al}_2\text{O}_3@LDGC)<0.001$; (c) Ti element. Statistical significance was analyzed using a t-test (n = 3). $P(\text{TiO}_2@LDGC\text{-TiO}_2/\text{Al}_2\text{O}_3@LDGC)<0.001$; (d) Al element. Statistical significance was analyzed using a t-test (n = 3). $P(\text{Al}_2\text{O}_3@LDGC\text{-TiO}_2/\text{Al}_2\text{O}_3@LDGC)<0.001$; Data are presented as mean \pm s.d. Source data are provided as a Source Data file." was added in the revised manuscript. (Page 22, Figure S11, Supplementary information)

10) Supplementary Figure 8 was added in the revised manuscript. (Page 19, Supplementary information)

Please find our responses to the comments and questions from the reviewers:

Reviewer #3 (Remarks to the Author):

Dear Authors, thank you very much for submitting your answers, interpretation and revised manuscript. Of course, you presented a lot of interesting results and a interesting method within your paper. But, still I don't belief that it is necessary to coat the surface of a dental restoration with a special nano-layer.

It is not discussed, that the effort to coat one or more restorations is really high and need educated persons. On the other hand you have to realize that after the process of fixation into the patients mooth the endodontics will be checked. That means, the dentist will remove some how the surface of a crown, inlay, onlay etc. Do you belive that the coated layer will be still there?

Please, you have to discuss this topics or make some remarks on that.

Response:

Thanks for your comment and suggestion. We fully understand and appreciate your concerns regarding the necessity of the nano-coating, the need for educated personnel for its practical implementation, and the potential impact of post-fixation adjustments on the coating's integrity. These are critical considerations for translating our technology into clinical practice, and we have carefully revised the manuscript to address each of these points in detail:

(1) But, still I don't belief that it is necessary to coat the surface of a dental restoration with a special nano-layer.

Answers:

Lithium silicate ceramics themselves have a bending strength of up to 450MPa, which can meet the requirements of dental standards. Due to their unique aesthetic advantages, they are widely used in clinical practice. However, there is still a certain degree of fracture rate in clinical use. Our data, as well as many existing studies, have revealed that in low-pH environments, ceramic surfaces will suffer from corrosion, and even surface structure defects can occur, thereby reducing their bending strength. The requirement in ISO standards to detect ion release also implies this issue, so it is

necessary to avoid this corrosion. Previous efforts have improved the bending strength of ceramics by adding zirconia crystals. Still, the quantity of added crystals is limited and cannot wholly prevent the corrosion process, which also increases cutting costs. Therefore, we propose using surface modification to avoid corrosion on ceramic surfaces directly.

Our work successfully coats the surface of lithium silicate ceramics with acid-resistant hybrid nanofilms using atomic layer deposition (ALD) technology. The thin films interact stably with the surface by chemical bonds and densely cover all complex morphologies, improving the acid and wear resistance without affecting the transparency of the ceramics.

(2) It is not discussed, that the effort to coat one or more restorations is really high and need educated persons.

Answers:

We understand your concerns regarding that the effort to coat one or more restorations is relatively high and requires highly educated personnel. We acknowledge that such expertise is indeed essential during the design phase of the ALD coating process. Since the ALD process is automated and controlled via computer programs, the procedure merely requires selecting the pre-configured program on a computer to initiate and complete the coating process once the deposition protocol is established. And the deposited thin film has advantages in highly controllable and uniform thickness, good conformality and high repeatability. Furthermore, the reaction chamber is capable of accommodating multiple restorations simultaneously for thin-film deposition. ALD is widely used in the fields like semiconductor. These advantages, along with amenability to automation, make it easy for non-professionals to perform operations.

(2) On the other hand you have to realize that after the process of fixation into the patients mooth the endodontics will be checked. That means, the dentist will remove some how the surface of a crown, inlay, onlay etc. Do you belive that the coated layer will be still there?

Answers:

On the other hand, the procedure for ALD coating in our work is as follows: first, the ceramic block is sectioned into the desired shape and trial-fitted on the patient's teeth. The occlusal height is adjusted to ensure optimal comfort. Subsequently, the ALD coating of Al₂O₃ and TiO₂ is conducted. Owing to the nanoscale thickness of the deposited film, no further adjustment of the coated ceramic is required, and it can be directly bonded and fixed in place, so there is no need to worry about any loss caused by checking.

Limitation:

The primary challenges hindering the widespread application of this technology are the development of low-cost ALD equipment, the optimization of deposition processes, and the successful completion of rigorous clinical testing and certification. Our research is actively focused on addressing these issues to facilitate the practical adoption and advancement of this technology.

The revisions list below aim to provide a more balanced assessment of our technology, acknowledging current limitations while emphasizing its potential value in specific clinical contexts.

Changes:

(1)"The flagship products, for example, IPS e.max CAD, have been fixtures in the dental market over 15 years."⁶ was modified as " The flagship products, for example, IPS e.max CAD with a flexural strength of up to 360 MPa,^{6, 7} have been fixtures in the dental market for over 15 years."⁸ (Page 2, in the first paragraph of Introduction)

(2)"Excessive addition of zirconia affects the transparency of the ceramic but increases the risk of crack generation during the cutting process."¹⁵⁻¹⁸ was modified as " For instance, the machinable zirconia-reinforced lithium silicate was developed under the models of VITA Suprinity and Celtra Duo, respectively.¹⁷ Their flexural strength can reach 451.4 MPa.¹⁸ Although the overall strength is improved, excessive addition of zirconia affects the transparency of the ceramic and even increases the risk of crack

generation during the cutting process.^{19-22"} (Page 2, in the first paragraph of Introduction)

(3)"Current research focuses mainly on enhancing the strength of dental ceramics, with limited attention given to their protection in harsh oral environments (e.g., acidic conditions). Our results show that even commercial crowns undergo acid corrosion (Supplementary Fig. 8), forming surface defects and pores that accelerate clinical failure and promote bacterial growth.^{24, 26, 43, 56-58} This work employs ALD for surface coating of lithium silicate glass ceramics with hybrid nanofilms. This method forms a dense hybrid oxide nanofilm through chemical bonding, which effectively blocks acid solution corrosion, prevents the formation of micropores and cracks, and preserves the original wear resistance of the glass ceramics without affecting their light transmittance. The pre-fabricated restorations first underwent occlusal height adjustment to ensure optimal comfort, followed by thin-film coating via ALD. Owing to the nanoscale thickness and high uniformity of the deposited film, no further adjustment of the coated ceramic is required. Furthermore, thanks to its capability for precise composition control, ALD technology holds promise for constructing thin films with tunable mechanical properties and good biocompatibility tailored to the specific mechanical requirements of different dental crowns. Moreover, the ALD process is automated and controlled via computer programs. Once the deposition protocol is established, the procedure merely requires selecting the pre-configured program on a computer to initiate and complete the coating process. Furthermore, the reaction chamber is capable of accommodating multiple restorations simultaneously for thin-film deposition. Thus, these advantages, along with amenability to automation, make it easy for non-professionals to perform operations. The primary direction for the development of this technology involves advancing low-cost, dedicated ALD equipment and achieving clinical certification." was added in the revised manuscript. (Page 19-20, in the last paragraph of Results and Discussion)

Reviewer #4 (Remarks to the Author):

The Reviewer's comments were addressed properly. The revision is adequate and I have no further comment to add; the ms can be accepted.

Response:

We are grateful to you for your positive assessment of our revisions.